# 🦚PEACOCK: MULTI-OBJECTIVE OPTIMIZATION FOR DEEP NEURAL NETWORK CALIBRATION

## ABSTRACT

The rapid adoption of deep neural networks underscores an urgent need for models to be safe, trustworthy and well-calibrated. Despite recent advancements in network calibration, the optimal combination of techniques remains relatively unexplored. By framing the task as a multi-objective optimization problem, we demonstrate that combining state-of-the-art methods can further boost calibration performance. We feature a total of seven state-of-the-art calibration algorithms and provide both theoretical and empirical motivation for their equal and weighted importance unification. We conduct experiments on both in-distribution and out-of-distribution computer vision and natural language benchmarks, investigating the speeds and contributions of different components. Our code is available anonymously at: *https://anonymous.4open.science/r/Peacock-1CE8* .

## 1 INTRODUCTION

Key requirements for the safe deployment of neural networks include multiple desirable qualities, such as high accuracy, fast training speeds and trustworthy predictions. While the recent successes in deep learning have increased the use of complex neural networks, a common observation is that deep models tend to be miscalibrated, exhibiting either under- or over-confident predictions (Guo et al., 2017).

Miscalibration can be particularly dangerous for high-stakes, safety-critical tasks such as medical prognosis (Esteva et al., 2017; Bandi et al., 2019), object-detection (Munir et al., 2023a;b; Liu et al., 2024), AI fairness and decision-making (Pleiss et al., 2017; Corvelo Benz & Rodriguez, 2023). Such tasks demand reliable decision-making algorithms, necessitating accurate confidence estimates that reflect a model's uncertainty (Jiang et al., 2018; Kendall & Gal, 2017). Specifically, calibration ensures that a model's predicted confidences align with its actual correctness. For instance, if a model assigns 0.9 confidence to a set of 100 samples, we should expect the model to be correct for 90 instances only.

Modern neural networks must not only remain well-calibrated in-distribution (ID), but also display invariance properties and remain robustly calibrated against out-of-distribution (OOD) shifts (Wald et al., 2021). This is crucial for real-world deployment, where models must generalize well and express uncertainty when handling unseen inputs (see Fig. 2b). For instance, OOD shifts in computer vision might involve changes in saturation and illumination, while in natural language tasks, they can arise from differences in syntax or spelling mistakes (Zhang et al., 2023).

While most calibration techniques tend to outperform the vanilla cross-entropy (CE) loss, they each tackle radically different issues, enabling independent performance boosts through different approaches. Each of these techniques exhibit varying trade-offs in ID/OOD performance (see Fig. 1 showing the 100% - ECE%)

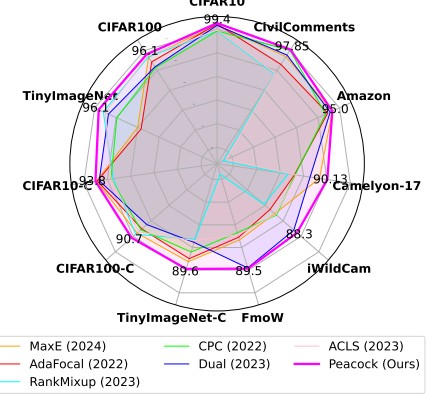

Figure 1: By unifying multiple calibration algorithms, *Peacock* outperforms individual methods, achieving state-of-the-art ID and OOD calibration. (Bigger is better)

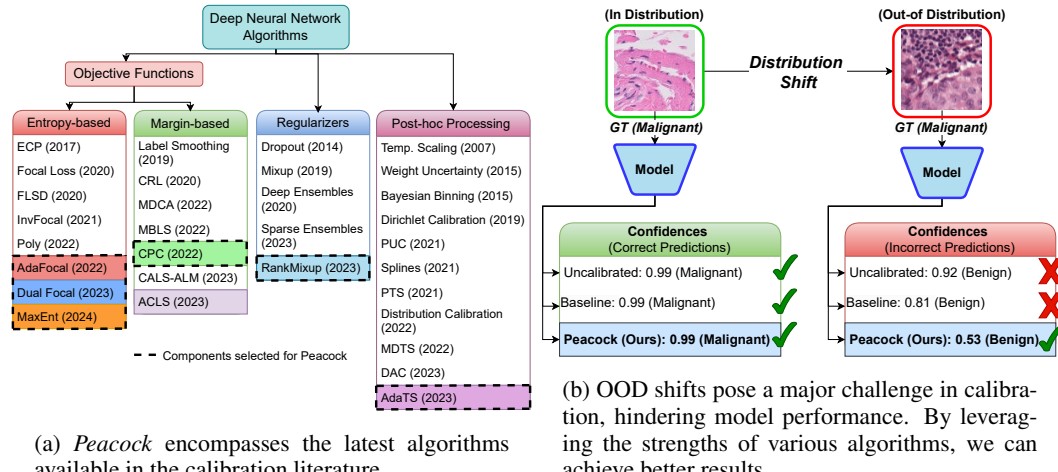

(a) *Peacock* encompasses the latest algorithms available in the calibration literature.

(b) OOD shifts pose a major challenge in calibration, hindering model performance. By leveraging the strengths of various algorithms, we can achieve better results.

Figure 2: Motivated by recent contributions, we propose *Peacock* for calibration under OOD shifts.

and computational complexity, making it difficult to pinpoint a clear winner (Cheng & Vasconcelos, 2022). Although it is theoretically possible for many calibration methods to be combined, the question remains on **which** of and **how** can these methods be fruitfully integrated together?

In this paper, our main goal is to *unify* different proposed calibration approaches into a single calibration framework named *Peacock*. Our claim is that by jointly optimizing multiple calibration objectives, performance boosts can be achieved for ID + OOD classification tasks. Theoretically, we demonstrate that *Peacock*'s calibration errors will always be bounded by the average calibration errors of all its components. We also propose a novel weighted importance form of *Peacock* that is fast and effective in balancing the contributions of different components. Apart from presenting *Peacock*, this paper is also doubly positioned as a literature survey of recently published calibration strategies (see Fig. 2a).

To demonstrate the combined efficacy of different calibration methods, we revisit a total of seven different SOTA calibration baselines and pick the final six to be integrated into *Peacock*. We further evaluate both ID and OOD performances of *Peacock* on popular synthetic and in-the-wild computer vision and natural language tasks. Our contributions can be summarized into the following points.

- **Peacock:** We present *Peacock*, a fully integrated multi-objective framework for deep neural network calibration.
- **Equal and Weighted Importance:** We motivate *Peacock* with theoretical guarantees and propose novel ways to weight the contributions of different calibration components.
- **Review of Literature:** This paper additionally serves as a condensed survey[1] of all existing algorithms proposed in the calibration literature.
- **Evaluation and Analysis:** We evaluate across popular synthetic and in-the-wild OOD vision and text benchmarks, empirically analyzing the speeds, effects and contributions of different components.

## 2 RELATED WORK

**Multi-Objective Optimization**   The primary goal of multi-objective optimization (MOO) is to simultaneously optimize multiple loss[2] terms $\{\mathcal{L}_1, ...\mathcal{L}_A\}$ on a single model. As these differing loss functions may possibly contrast and conflict with each other (Sener & Koltun, 2018). Obtaining a way to balance/weight these losses during optimization is of great interest, since learned representations between losses can be shared (Caruana, 1997; Zamir et al., 2018), with the added benefit

---

[1] To the best of our ability, we cover all published works and further discuss them in Appendix A

[2] For clarity, the term **loss** loosely refers to auxiliary functions, whilst **objective function** represents the final learning goal during model training.

of avoiding model redundancy in the form of large ensembles (Dosovitskiy & Djolonga, 2020). Prevalent methods include grid-search tuning, which can be cumbersome and fixating predefined weights during training may not guarantee optimal performance (Groenendijk et al., 2021). Other approaches include learning the hardest task first (Guo et al., 2018), self-paced learning (Li et al., 2017), aleatoric uncertainty estimation (Kendall & Gal, 2017; Kendall et al., 2018), gradient normalization (Chen et al., 2018), pareto frontiers (Sener & Koltun, 2018; Lin et al., 2019; Xiao et al., 2023; Liu et al., 2021) and co-efficient of variations (Groenendijk et al., 2021). Additionally, we refer readers to (Zhang & Yang, 2017; Gong et al., 2019) for a comprehensive review and comparisons on multi-objective methods. In this work, our focus is directed towards gradient-based multi-objective optimization for balancing calibration loss terms.

**Deep Uncertainty Calibration** In Fig. 2a, we provide an overview of recent calibration algorithms and metrics. Examples of these algorithms include (1) *Entropy-based* methods that control the entropy of the model (Mukhoti et al., 2020; Wang et al., 2021; Leng et al., 2022; Ghosh et al., 2022; Tao et al., 2023; Neo et al., 2024). (2) *Margin-based* methods that directly limit model confidences (Hebbalaguppe et al., 2022; Liu et al., 2022a; Cheng & Vasconcelos, 2022; Liu et al., 2023a;b) (3) *Regularizers* that augment the training inputs or model (Zhang et al., 2020; Sapkota et al., 2023; Noh et al., 2023). (4) *Post-hoc processing*, which requires tuning the model on a hold-out validation set in order to scale predictions (Wenger et al., 2020; Tomani et al., 2021; Gupta et al., 2021; Tomani et al., 2022; Kuleshov & Deshpande, 2022; Gruber & Buettner, 2022; Yu et al., 2022; Tomani et al., 2023; Joy et al., 2023). Calibration metrics include binning and binning-free approaches (Gupta et al., 2021; Roelofs et al., 2022; Yang et al., 2023; Xiong et al., 2023). Additionally, we refer readers to Appendix A.5 for a discussion on calibration metrics. To keep the scale of our experiments manageable, we highlight only the latest algorithms of each sub-group used in *Peacock* (e.g., AdaFocal).

# 3 BACKGROUND

## 3.1 DEEP NEURAL NETWORK CALIBRATION

Consider a classification problem over an input feature space $X$ and output space $Y$, where $N$ labelled i.i.d pairs $(x_i, y_i)_{i=1}^{N}$ are randomly sampled from a training set $\mathcal{D}$. The model/hypothesis is then simply a mapping $h_{\boldsymbol{\theta}} : X \to Y$, where $Y \in [0, 1]$ and $\boldsymbol{\theta}$ denotes a deep neural network consisting of $K$ neurons. The model is tasked to estimate a valid posterior such that $\sum_{k=1}^{K} P_i(y_k|x) = 1$, with the predicted top-1 class label $\hat{y} := \arg\max h_i^{\boldsymbol{\theta}}(x)$ obtained from the logits with the top softmax confidence $\hat{P}(h^{\boldsymbol{\theta}}) := \max_k P_i(y_k|x)$. The model is considered *perfectly calibrated* if and only if its confidence matches its probability of being correct, satisfying the formal definition $\mathbb{P}(\hat{y} = y|\hat{P} = P) = P \quad \forall \in P[0 - 1]$. As this definition of calibration cannot be computed with finite samples, the most widely used approximation is the expected calibration error (ECE) (Naeini et al., 2015):

**Definition 3.1 (Expected Calibration Error)** *The empirical expected calibration error of a single hypothesis $h^{\boldsymbol{\theta}}(x)$ can be written as Eq.3 in (Zhang et al., 2020) and Eq.7 in (Yang et al., 2023):*

$$\text{ECE}^d(h^{\boldsymbol{\theta}}) = \sum_{b=1}^{B} \frac{n_b}{N} ||\bar{P}(h_b^{\boldsymbol{\theta}}(x)) - \bar{y}_b||_d^d \tag{1}$$

whereby the average predicted confidences $\bar{P}(h_b^{\boldsymbol{\theta}}(x))$ and targets $\bar{y}_b$ are partitioned into $B$ bins, each containing $n_b$ samples and $||.||_d^d$ is the $d$-th power of the $\mathcal{L}_d$ norm between the predictions and targets.

For OOD scenarios, the test distribution may diverge from the samples observed during training. Specifically, these OOD shifts can be caused by either concept shifts to the classes (changes in the posterior distribution $P(Y|X)$) or covariate shifts to input features (changes in the marginal distribution $P(X)$) (Shen et al., 2021). These OOD shifts tend to degrade model accuracy and calibration (Ovadia et al., 2019) which can be problematic for deployment. Unfortunately, achieving good calibration on both ID and OOD problem sets is non-trivial, since OOD samples typically vary

greatly from ID samples with the type and magnitude of shift unknown (Neo et al., 2024). In this work, our focus is on the problem of covariate shifts, with the goal of achieving good top-1 calibration and generalization across both ID and OOD settings.

## 3.2 Calibration Algorithms and Techniques

Although many calibration algorithms have been proposed, each of these works tackle fundamentally different issues, with varying results and no consensus on which approach is the best. We diagnose the prevalent issues in deep neural network calibration and highlight the approaches of seven SOTA algorithms.

**Adaptive Focal Parameter Selection**  The Focal loss (FL) (Lin et al., 2017) has been a pivotal contribution in network calibration (Mukhoti et al., 2020). As a trade-off between minimizing the Kullback-Leibler divergence and maximizing entropy, the FL: $\mathcal{L}_F = -\sum_k \left(1 - P_i(y_k|x_i)\right)^\gamma \log P_i(y_k|x_i)$ is sensitive to the hyper-parameter $\gamma$, which controls the convexity of the entropy term. While strictly setting $\gamma > 1$ reduces over-confidence, it can also cause under-confidence. To circumvent this, Adaptive FL (AdaFocal) (Ghosh et al., 2022) conditionally switches between the FL and Inverse FL (Wang et al., 2021) with different selected values of $\gamma$.

$$\mathcal{L}_{\text{Ada}} = \begin{cases} -\sum_k \left(1 - P_i(y_k|x)\right)^{\gamma_{t,b}} \log P_i(y_k|x) & \text{if } \gamma_{t,b} \geq 0 \\ -\sum_k \left(1 + P_i(y_k|x)\right)^{|\gamma_{t,b}|} \log P_i(y_k|x) & \text{if } \gamma_{t,b} < 0, \end{cases} \tag{2}$$

**Maximum Entropy Constraints**  Based on the Principle of Maximum Entropy (Jaynes, 1957) and an extension of the FL. MaxEnt loss (Neo et al., 2024) is designed to handle OOD samples using statistical constraints computed from the prior distribution of the training set.

$$\mathcal{L}_M^{ME} = -\sum_k (1 - P_i(y_k|x))^\gamma \log P_i(y_k|x)$$

$$+ \lambda_\mu \underbrace{\left[\sum_k f(\mathcal{Y})P_i(y_k|x) - \mu_G\right.}_{\text{Global Expectation}} + \underbrace{\left.\sum_k f(\mathcal{Y})P_i(y_k|x) - \mu_{Lk}\right]}_{\text{Local Expectation}} \tag{3}$$

Whereby the global expectations are computed from the entire training set such as $\mathbb{E}[\mathcal{Y}] = \sum_k P_i(y_k|x)f(\mathcal{Y}) = \mu_G$ and the local expectations are computed sample-wise from the class value characteristic function $f(\mathcal{Y})$. The Lagrange multiplier $\lambda_\mu$ controls the strength of the constraints, which can be solved cheaply using a numerical root-finder.

**Under- and Over-confidence Trade-off**  A caveat to FL and its extensions alike, is that maximizing the entropy term tends to penalize all output predictions, causing under-confidence (Charoenphakdee et al., 2021). Dual FL (Tao et al., 2023) maximizes the gap between the ground truth $P_i(y_{GT}|x)$ and the highest confidence $P_i(y_j|x)$ after the $\arg\max$ class, balancing the trade-off between over- and under-confident predictions.

$$\mathcal{L}_{\text{Dual}} = -\sum_k (1 - P_i(y_k|x) + P_i(y_j|x))^\gamma \log P_i(y_k|x)$$

$$\text{where} \quad P_i(y_j|x) = \max_k \{P_i(y_k|x) | P_i(y_k|x) < P_i(y_{GT}|x)\} \tag{4}$$

**Pairwise Binary Discriminatory Constraints**  As binary problems are easier to calibrate, CPC loss (Cheng & Vasconcelos, 2022) proposes to decompose the original multi-class problem into $\frac{K(K-1)}{2}$ binary classification problems. Whereby the predictions $P_i(y_k|x)$ are calibrated against the confidences $P_i(y_l|x)$ of the remaining $(K - 1)$ pairs that do not involve the true class:

$$\mathcal{L}_{\text{CPC}}^{\text{1v1}} = -\frac{1}{(K-1)} \sum_{l \neq k} \log \frac{P_i(y_k|x)}{P_i(y_k|x) + P_i(y_l|x)} \quad (5)$$

**Conditional Label Smoothing**   Label smoothing (LS) (Müller et al., 2019) improves calibration by artificially softening targets with a constant margin $\epsilon$. However, LS often leads to under-confident predictions and requires time-consuming grid searches to find an optimal $\epsilon$. To address these limitations, several approaches have proposed adaptive or conditional label smoothing functions (see Appendix A). Building upon these methods, Adaptive Conditional Label Smoothing (ACLS) (Park et al., 2023) aims to dynamically approximate the label smoothing function.

$$\mathcal{L}_{\text{ACLS}} = \begin{cases} \lambda_1 \max(0, h_k^\theta(x) - \min_k(h_k^\theta(x)) - m_{\text{ACLS}})^2 & \text{if } k = \hat{y} \\ \lambda_2 \max(0, h_{\hat{y}}^\theta(x) - h_k^\theta(x) - m_{\text{ACLS}})^2 & \text{if } k \neq \hat{y} \end{cases} \quad (6)$$

When $k = \hat{y}$, the smoothing function is directly proportional to $h_k^\theta(x)$, thereby lowering confidences. Similarly, when $k \neq \hat{y}$, the effects of the smoothing function decreases, allowing the logits and confidences to increase. $m_{\text{ACLS}}$ denotes the ACLS margin and $\lambda_1$, $\lambda_2$ are hyperparameters for cases when $k = \hat{y}$ and $k \neq \hat{y}$.

**Feature and Label Regularization**   Mixup (Zhang et al., 2018) is highly effective for network calibration (Thulasidasan et al., 2019; Chidambaram & Ge, 2024; Zhang et al., 2022). By interpolating a pair of inputs $(x_i, x_j)$ and targets $(y_i, y_j)$, the augmented inputs and smoothed labels $(\tilde{x}, \tilde{y})$ are obtained using the following equations:

$$\begin{aligned} \tilde{x} &= \beta x_i + (1 - \beta)x_j \\ \tilde{y} &= \beta y_i + (1 - \beta)y_j \end{aligned} \quad (7)$$

where $\beta \in [0-1] \sim \text{Beta}(\alpha, \alpha)$ is a blending coefficient, randomly drawn from a Beta distribution. By considering the ordinal ranking of training samples, RankMixup (Noh et al., 2023) further improves vanilla mixup by enforcing the confidences of interpolated samples to be lower than the confidences of original samples. The ordinal relationship between "easy" and "hard" samples is maintained by a margin $m_{\text{MRL}}$.

$$\mathcal{L}_{\text{MRL}} = \max(0, \max_k \tilde{P}_i(\tilde{y}|\tilde{x}) - \max_k P_i(y|x) + m_{\text{MRL}}) \quad (8)$$

RankMixup (Noh et al., 2023) can be computationally inefficient due to its requirement for two forward passes: one for the original samples $P_i(y|x)$ and another for the mixed samples $\tilde{P}_i(\tilde{y}|\tilde{x})$. To improve computational efficiency, we propose an optimized version of RankMixup within *Peacock* that performs image and label mixing *batchwise*, enabling a single forward pass and faster compute times for $\tilde{P}_i(\tilde{y}|\tilde{x})$. Additional details for speeding up RankMixup can be found in Appendix B.2.

**Adaptive Temperature Scaling**   As a post-hoc method, temperature scaling (TS) (Platt & Karampatziakis, 2007) manipulates the predictions by a scalar $\mathcal{T} \in \mathcal{R}^+$. Similar to LS, TS tends to reduce the confidence of every sample - even for correct predictions and finding a suitable $\mathcal{T}$ requires a grid-search over a separate validation set. Adaptive Temperature scaling (AdaTS) (Joy et al., 2023) aims to learn samplewise temperatures from the features $h^\theta(x)$. By jointly learning a conditional variational autoencoder (Kingma & Welling, 2014) and a multi-layer perceptron $\phi$, the samplewise temperatures are obtained as a post-processing step.

$$\mathcal{L}_{\text{AdaTS}} = -\mathbb{ELBO}[h_i^\theta(x)] - \log\left(\frac{\exp\left(h_i^\theta(x)/\mathcal{T}_i\right)}{\sum_{k=1}^K \exp\left(h_k^\theta(x)/\mathcal{T}_i\right)}\right) \quad (9)$$

## 4   MOTIVATION AND PEACOCK (PUTTING IT ALL TOGETHER)

**Component Synergy**   Why combine calibration methods? Despite their diverse approaches, all calibration algorithms discussed in Section 3.2 share the common goal of enhancing model cali-

bration. This suggests that they can be effectively integrated into a unified framework, leveraging their complementary strengths to achieve even better results. This section outlines our theoretical motivation for unifying calibration algorithms into *Peacock*. We first demonstrate how their equal combination improves calibration performance, subsequently we propose a novel weighted importance formulation that dynamically balances loss terms to further boost performance.

## 4.1 EQUAL IMPORTANCE FORMULATION

Consider a multi-objective function $\mathcal{L}(\boldsymbol{\theta}) = \frac{1}{A} \sum_{t=1}^{A} \mathcal{L}_t(\boldsymbol{\theta})$ comprising of a linear, equally weighted sum of $A$ correlated loss terms/algorithms. From Definition 3.1, each empirical loss term $\mathcal{L}_t(\boldsymbol{\theta}) \triangleq \frac{1}{N} \sum_i \mathcal{L}(\hat{P}_i(h_t^{\boldsymbol{\theta}}), y_i))$ yields an individual hypothesis $\hat{P}(h_t^{\boldsymbol{\theta}})$ with a corresponding calibration error $\hat{P}(h_t^{\boldsymbol{\theta}}) = \bar{y} + \text{ECE}^d(h_t^{\boldsymbol{\theta}})$. The unified hypothesis $H^{\boldsymbol{\theta}}$ of the multi-objective learner $\mathcal{L}(\boldsymbol{\theta})$ can then be interpreted as the average of each individual hypothesis $\bar{P}(H^{\boldsymbol{\theta}}) = \frac{1}{A} \sum_{t=1}^{A} \hat{P}(h_t^{\boldsymbol{\theta}}) = \bar{y} + \text{ECE}^d(H^{\boldsymbol{\theta}})$. When $d = 2$, the averaged squared ECE across all individual hypotheses is given by:

$$\overline{\text{ECE}}^2(h^{\boldsymbol{\theta}}) = \frac{1}{A} \sum_{t=1}^{A} \text{ECE}^2(h_t^{\boldsymbol{\theta}}) = \frac{\text{ECE}(h_1^{\boldsymbol{\theta}})^2 + \text{ECE}(h_2^{\boldsymbol{\theta}})^2 + ... + \text{ECE}(h_t^{\boldsymbol{\theta}})^2}{A} \tag{10}$$

As we equally consider the contributions of each individual hypotheses/loss term, with some rearrangement the expected squared ECE of the unified multi-objective learner can be obtained as:

$$\text{ECE}^2(H^{\boldsymbol{\theta}}) = \mathbb{E}_x\left[\left(\frac{1}{A} \sum_{t=1}^{A} \text{ECE}(h_t^{\boldsymbol{\theta}}(x))\right)^2\right] = \int \left(\frac{1}{A} \sum_{t=1}^{A} \text{ECE}(h_t^{\boldsymbol{\theta}}(x))\right)^2 p(x)dx \tag{11}$$

where $p(x)$ is the prior probability of each input. Then from Eq. (10) and Eq. (11), the combined learner is safely bounded by the averaged squared ECE of all individual hypotheses.

$$\text{ECE}^2(H^{\boldsymbol{\theta}}) \le \overline{\text{ECE}}^2(h^{\boldsymbol{\theta}}) \tag{12}$$

As the $\text{ECE}^1$ and $\text{ECE}^2$ are highly correlated (Zhang et al., 2020), we expect the upper bound in Eq. (12) to hold. This upper bound remains applicable even for temperature-scaled variants of each hypothesis, where $\mathcal{T}$ is a temperature function.

$$\text{ECE}^2(\mathcal{T}(H^{\boldsymbol{\theta}})) \le \overline{\text{ECE}}^2(\mathcal{T}(h^{\boldsymbol{\theta}})) \tag{13}$$

*Proof.* See Appendix C.

Similar to model ensembles (Zhou, 2012), loss ensembles allows a single model to perform well on multiple tasks, with additional practical benefits, such as sharing lower-level features and better compute times (Dosovitskiy & Djolonga, 2020). However, loss terms can often be conflicting, requiring trade-offs between different objectives.

## 4.2 WEIGHTED IMPORTANCE FORMULATION

To address these challenges, we propose a weighted importance formulation in the following section. This approach aims to find a suitable set of weights that optimizes the overall performance of the multi-objective learner. The weighted multi-objective optimization problem generally yields the following minimization problem:

$$\min_{\boldsymbol{\theta}} \mathcal{L}(\boldsymbol{\theta}) = \sum_{t=1}^{A} w_t \mathcal{L}_t(\boldsymbol{\theta}) \tag{14}$$

where $w_t$ are a set of unknown scalar weights controlling each loss term. In many cases, obtaining a suitable set of weights for Eq. (14) is highly desirable. However, common approaches would either typically require expensive grid searches or predefined heuristics (Kendall et al., 2018; Chen et al., 2018). A well-studied approach in multi-objective optimization are Pareto optimal solutions, which delivers different trade-offs amongst loss terms. The goal of achieving Pareto optimality (Sener & Koltun, 2018; Lin et al., 2019) is defined with the following necessary conditions.

**Definition 4.1** *(Conditions for Pareto Optimal Calibration)*

    1. **Pareto dominance** *A solution $\boldsymbol{\theta}$ dominates another solution $\bar{\boldsymbol{\theta}}$ where $\boldsymbol{\theta} \prec \bar{\boldsymbol{\theta}}$, if $\mathcal{L}^t(\boldsymbol{\theta}) \leq \mathcal{L}^t(\bar{\boldsymbol{\theta}})$ for all objectives $t$ and $\mathbf{L}(\boldsymbol{\theta}_1, ..., \boldsymbol{\theta}_A) \neq \mathbf{L}(\bar{\boldsymbol{\theta}}_1, ..., \bar{\boldsymbol{\theta}}_A)$).*

    2. **Pareto optimality** *Solution $\boldsymbol{\theta}^*$ is considered Pareto optimal if there exists no other solution $\boldsymbol{\theta}$ that dominates $\boldsymbol{\theta}^*$ such that $\boldsymbol{\theta} \prec \boldsymbol{\theta}^*$.*

Assuming loss terms are convex and optimizable with gradient descent, Pareto optimal weights for each loss can be obtained through the Karush-Kuhn-Tucker (KKT) conditions (Fliege & Svaiter, 2000; Schäffler et al., 2002), by minimizing the following objective (Désidéri, 2012; Sener & Koltun, 2018):

$$\min_{w_1,...,w_t} \left\{ \left\| \sum_{t=1}^{A} w_t \nabla_{\boldsymbol{\theta}} \mathcal{L}_t(\boldsymbol{\theta}) \right\|_2^2 \, \middle| \, \sum_{t=1}^{A} w_t = 1, w_t \geq 0 \quad \forall t \right\} \tag{15}$$

Previous works (Désidéri, 2012; Sener & Koltun, 2018) have shown that the solution to Eq. (15) is either zero or provides a gradient direction that improves all loss components.

**Practical Considerations** Generally, optimizing for $w_t$ in Eq. (15) requires a separate optimizer and the recomputation of the gradients $\nabla_{\boldsymbol{\theta}} \mathcal{L}_t(\boldsymbol{\theta})$ for each loss term. This involves retaining the computational graph[3] for $A$ backward passes, which slows down training speeds and grows prohibitively more expensive as $A$ becomes larger.

To circumvent this, we propose a fast, elegant and efficient alternative to recomputing gradients, by replacing $\nabla_{\boldsymbol{\theta}} \mathcal{L}_t(\boldsymbol{\theta})$ with decrease rate estimates for each loss term. Assuming model parameters $\boldsymbol{\theta}$ are updated via $\boldsymbol{\theta}' \leftarrow \boldsymbol{\theta} - \eta \nabla_{\boldsymbol{\theta}} \mathcal{L}_t(\boldsymbol{\theta})$, we propose to balance learning loss terms with the following direction-orientated objective:

$$\min_{w_1,...,w_t} \left\{ \left\| \sum_{t=1}^{A} w_t \sqrt{\frac{\Delta_{\boldsymbol{\theta}} \mathcal{L}_t(\boldsymbol{\theta})}{\eta}} \right\|_2^2 \, \middle| \, \sum_{t=1}^{A} w_t = 1, w_t \geq 0 \quad \forall t \right\} \tag{16}$$

*Proof.* See Appendix C.2.

where the decrease rate estimates $\sqrt{\frac{\Delta_{\boldsymbol{\theta}} \mathcal{L}_t(\boldsymbol{\theta})}{\eta}}$, are derived using the first-order Taylor approximation with a sufficiently small step size $\eta$. As long as the KKT conditions are satisfied and loss terms $\mathcal{L}_t(\boldsymbol{\theta})$ are monotonically decreasing, faster performance can be achieved using Eq. (16). With the added benefit of $w_t \propto \sqrt{\frac{\Delta_{\boldsymbol{\theta}} \mathcal{L}_t(\boldsymbol{\theta})}{\eta}}$ which ensures balanced learning rates across all loss terms, preventing any single term from dominating the optimization process.

**Direction Weighted Self-Attention** To optimize the objective in Eq. (16), we propose using a direction weighted self-attention block. Fig. 3 illustrates our self-attention block, which accepts an array of loss terms $\mathcal{L}_t(\boldsymbol{\theta})$ as inputs and outputs a set of weights $w_t$. The Value (V), Key (K) and Query (Q) neurons are of size $A \times A$ and the softmax function $\sigma$ is applied to ensure that $\sum_t w_t = 1$. The learning dynamics of the direction weighted self-attention block are discussed in Appendix B.3.

Full details can be found in Algorithm 1, which shows all calibration components of *Peacock* and an optional step for obtaining importance weights. A peculiar finding in our ablation study, is that removing ACLS tends to lead to better performance in *Peacock*. Since $\mathcal{L}_{\text{AdaFocal}}^{\text{Dual}}$ contains the CE loss, we only apply importance weights to the auxiliary loss terms with the final objective function given by: $\mathcal{L}^{\text{Peacock}} = \mathcal{L}_{\text{AdaFocal}}^{\text{Dual}} + w_1 \mathcal{L}_{\text{constraints}}^{ME} + w_2 \mathcal{L}_{\text{CPC}}^{1v1} + w_3 \mathcal{L}_{\text{MRL}}$.

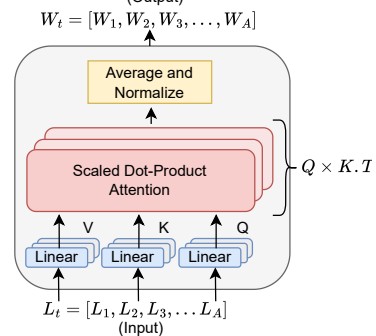

Figure 3: Our direction weighted self-attention block learns the importance of each loss term.

---

[3] For more details, see Pytorch autograd framework: *https://pytorch.org/docs/stable/autograd.html*

| Dataset | Metric | CE | MaxEnt | AdaFocal | RankMixup | CPC | Dual | ACLS | Peacock (Eq.) | Peacock (Impt.) |
|---|---|---|---|---|---|---|---|---|---|---|
| CIFAR10-C | Acc. ↑ | $77.9_{\pm0.3}$ | $78.3_{\pm0.2}$ | $77.7_{\pm0.3}$ | $77.8_{\pm0.3}$ | $77.6_{\pm0.2}$ | $77.9_{\pm0.3}$ | $78.1_{\pm0.4}$ | $76.8_{\pm0.2}$ | $77.3_{\pm0.4}$ |
| | ECE ↓ | $14.5_{\pm0.4}$ | $6.5_{\pm0.2}$ | $6.8_{\pm0.2}$ | $11.9_{\pm0.3}$ | $11.2_{\pm0.2}$ | $7.4_{\pm0.3}$ | $11.2_{\pm0.4}$ | $\underline{6.3_{\pm0.1}}$ | $\mathbf{6.2_{\pm0.3}}$ |
| | CECE ↓ | $3.3_{\pm0.1}$ | $2.6_{\pm0.1}$ | $2.6_{\pm0.1}$ | $2.9_{\pm0.1}$ | $2.8_{\pm0.1}$ | $2.7_{\pm0.1}$ | $2.8_{\pm0.1}$ | $\underline{2.7_{\pm0.1}}$ | $\mathbf{2.6_{\pm0.1}}$ |
| | KSE ↓ | $14.5_{\pm0.4}$ | $6.2_{\pm0.1}$ | $6.5_{\pm0.3}$ | $11.8_{\pm0.2}$ | $11.1_{\pm0.1}$ | $7.0_{\pm0.3}$ | $11.3_{\pm0.4}$ | $\mathbf{6.1_{\pm0.2}}$ | $\mathbf{6.1_{\pm0.3}}$ |
| CIFAR100-C | Acc. ↑ | $52.5_{\pm0.1}$ | $52.4_{\pm0.1}$ | $52.9_{\pm0.2}$ | $52.3_{\pm0.1}$ | $52.0_{\pm0.1}$ | $52.8_{\pm0.1}$ | $52.6_{\pm0.1}$ | $51.8_{\pm0.1}$ | $52.6_{\pm0.1}$ |
| | ECE ↓ | $10.6_{\pm0.1}$ | $11.6_{\pm0.6}$ | $13.6_{\pm0.1}$ | $11.0_{\pm1.4}$ | $13.2_{\pm0.5}$ | $15.7_{\pm0.1}$ | $12.2_{\pm0.2}$ | $\underline{9.6_{\pm0.2}}$ | $\mathbf{9.3_{\pm0.1}}$ |
| | CECE ↓ | $0.4_{\pm0.1}$ | $0.5_{\pm0.1}$ | $0.5_{\pm0.1}$ | $0.4_{\pm0.1}$ | $0.4_{\pm0.1}$ | $0.5_{\pm0.1}$ | $0.4_{\pm0.1}$ | $\mathbf{0.4_{\pm0.1}}$ | $\mathbf{0.4_{\pm0.1}}$ |
| | KSE ↓ | $9.3_{\pm0.1}$ | $11.4_{\pm0.6}$ | $13.3_{\pm0.1}$ | $10.3_{\pm1.5}$ | $11.5_{\pm0.5}$ | $15.3_{\pm0.1}$ | $11.7_{\pm0.3}$ | $\underline{9.6_{\pm0.4}}$ | $\mathbf{9.2_{\pm0.6}}$ |
| TinyImageNet-C | Acc. ↑ | $25.2_{\pm0.1}$ | $22.0_{\pm0.1}$ | $25.1_{\pm0.1}$ | $23.1_{\pm0.3}$ | $23.7_{\pm0.1}$ | $22.9_{\pm0.6}$ | $22.1_{\pm0.1}$ | $23.3_{\pm0.5}$ | $23.6_{\pm0.2}$ |
| | ECE ↓ | $15.7_{\pm0.5}$ | $12.8_{\pm0.1}$ | $13.8_{\pm0.4}$ | $20.2_{\pm0.1}$ | $16.0_{\pm0.5}$ | $19.2_{\pm0.4}$ | $19.8_{\pm0.2}$ | $\underline{10.6_{\pm0.2}}$ | $\mathbf{10.4_{\pm0.2}}$ |
| | CECE ↓ | $0.3_{\pm0.1}$ | $0.3_{\pm0.1}$ | $0.3_{\pm0.1}$ | $0.4_{\pm0.1}$ | $0.3_{\pm0.1}$ | $0.3_{\pm0.1}$ | $0.3_{\pm0.1}$ | $\underline{0.3_{\pm0.1}}$ | $\mathbf{0.3_{\pm0.1}}$ |
| | KSE ↓ | $15.7_{\pm0.5}$ | $12.8_{\pm0.2}$ | $13.8_{\pm0.3}$ | $20.2_{\pm0.2}$ | $15.7_{\pm0.2}$ | $19.2_{\pm0.7}$ | $19.8_{\pm0.2}$ | $\underline{10.6_{\pm0.2}}$ | $\mathbf{10.3_{\pm0.2}}$ |
| Camelyon17 | Acc. ↑ | $81.7_{\pm0.7}$ | $79.7_{\pm1.7}$ | $74.5_{\pm1.7}$ | $74.9_{\pm4.0}$ | $77.7_{\pm1.6}$ | $78.4_{\pm2.9}$ | $77.0_{\pm1.2}$ | $79.3_{\pm2.7}$ | $83.2_{\pm1.1}$ |
| | ECE ↓ | $15.5_{\pm1.1}$ | $12.4_{\pm0.1}$ | $20.4_{\pm0.4}$ | $22.4_{\pm4.6}$ | $20.2_{\pm1.6}$ | $15.4_{\pm2.5}$ | $19.8_{\pm0.2}$ | $\underline{11.7_{\pm0.7}}$ | $\mathbf{9.8_{\pm1.8}}$ |
| | CECE ↓ | $16.7_{\pm1.3}$ | $16.2_{\pm0.1}$ | $23.7_{\pm0.7}$ | $23.6_{\pm4.8}$ | $21.3_{\pm2.0}$ | $20.3_{\pm2.9}$ | $22.0_{\pm1.1}$ | $\underline{14.0_{\pm0.4}}$ | $\mathbf{13.7_{\pm1.6}}$ |
| | KSE ↓ | $15.5_{\pm1.1}$ | $12.3_{\pm0.8}$ | $20.4_{\pm0.4}$ | $22.4_{\pm4.6}$ | $20.2_{\pm1.6}$ | $15.4_{\pm2.4}$ | $19.6_{\pm0.1}$ | $\underline{11.7_{\pm0.7}}$ | $\mathbf{9.8_{\pm1.8}}$ |
| iWildCam | Acc. ↑ | $52.2_{\pm0.3}$ | $50.9_{\pm1.0}$ | $54.1_{\pm1.7}$ | $56.7_{\pm0.3}$ | $55.8_{\pm2.2}$ | $54.6_{\pm2.1}$ | $55.2_{\pm2.2}$ | $51.7_{\pm0.8}$ | $54.5_{\pm0.8}$ |
| | ECE ↓ | $30.6_{\pm0.8}$ | $21.0_{\pm3.2}$ | $23.0_{\pm0.5}$ | $25.5_{\pm0.7}$ | $20.3_{\pm1.1}$ | $13.0_{\pm2.5}$ | $20.6_{\pm1.8}$ | $\mathbf{9.7_{\pm0.3}}$ | $\underline{12.6_{\pm1.4}}$ |
| | CECE ↓ | $0.4_{\pm0.1}$ | $0.4_{\pm0.1}$ | $0.3_{\pm0.1}$ | $0.4_{\pm0.1}$ | $0.3_{\pm0.1}$ | $0.4_{\pm0.1}$ | $0.3_{\pm0.1}$ | $\mathbf{0.3_{\pm0.1}}$ | $\underline{0.3_{\pm0.1}}$ |
| | KSE ↓ | $30.6_{\pm0.8}$ | $21.0_{\pm3.2}$ | $23.0_{\pm0.5}$ | $25.5_{\pm0.7}$ | $19.5_{\pm1.4}$ | $13.0_{\pm2.5}$ | $20.6_{\pm1.8}$ | $\mathbf{9.7_{\pm0.3}}$ | $\underline{12.6_{\pm1.4}}$ |
| FmoW | Acc. ↑ | $35.1_{\pm0.5}$ | $33.5_{\pm0.1}$ | $35.5_{\pm0.7}$ | $35.8_{\pm0.1}$ | $36.4_{\pm0.1}$ | $35.1_{\pm0.2}$ | $37.5_{\pm0.1}$ | $35.1_{\pm0.2}$ | $35.5_{\pm0.1}$ |
| | ECE ↓ | $39.8_{\pm0.2}$ | $20.0_{\pm9.9}$ | $20.9_{\pm8.6}$ | $41.7_{\pm0.1}$ | $22.4_{\pm0.9}$ | $10.7_{\pm0.1}$ | $21.7_{\pm0.2}$ | $\underline{10.6_{\pm0.4}}$ | $\mathbf{10.5_{\pm0.3}}$ |
| | CECE ↓ | $1.5_{\pm0.1}$ | $1.0_{\pm0.3}$ | $1.0_{\pm0.2}$ | $1.5_{\pm0.1}$ | $0.9_{\pm0.1}$ | $0.6_{\pm0.1}$ | $0.9_{\pm0.1}$ | $\underline{0.6_{\pm0.1}}$ | $\mathbf{0.6_{\pm0.1}}$ |
| | KSE ↓ | $39.8_{\pm0.2}$ | $20.0_{\pm9.9}$ | $20.9_{\pm8.6}$ | $41.7_{\pm0.1}$ | $22.4_{\pm0.9}$ | $10.7_{\pm0.1}$ | $21.7_{\pm0.2}$ | $\underline{10.6_{\pm0.4}}$ | $\mathbf{10.5_{\pm0.3}}$ |
| Amazon | Acc. ↑ | $55.8_{\pm0.3}$ | $64.6_{\pm0.4}$ | $59.6_{\pm2.7}$ | $56.9_{\pm0.1}$ | $56.9_{\pm0.1}$ | $60.7_{\pm3.8}$ | $56.9_{\pm0.2}$ | $57.5_{\pm0.6}$ | $64.9_{\pm0.8}$ |
| | ECE ↓ | $7.0_{\pm0.5}$ | $\underline{5.0_{\pm0.6}}$ | $6.7_{\pm1.0}$ | $43.1_{\pm0.1}$ | $7.4_{\pm0.5}$ | $5.8_{\pm2.3}$ | $42.0_{\pm0.1}$ | $6.5_{\pm3.2}$ | $\mathbf{5.0_{\pm1.1}}$ |
| | CECE ↓ | $6.4_{\pm0.3}$ | $3.8_{\pm0.6}$ | $3.3_{\pm0.8}$ | $17.2_{\pm0.1}$ | $4.8_{\pm0.2}$ | $2.5_{\pm0.9}$ | $16.8_{\pm0.1}$ | $\underline{2.9_{\pm1.3}}$ | $\mathbf{2.3_{\pm0.3}}$ |
| | KSE ↓ | $7.0_{\pm0.5}$ | $\mathbf{5.1_{\pm0.6}}$ | $7.5_{\pm0.6}$ | $43.1_{\pm0.3}$ | $10.9_{\pm3.3}$ | $8.1_{\pm0.1}$ | $42.1_{\pm0.1}$ | $8.6_{\pm1.0}$ | $\underline{6.6_{\pm0.4}}$ |
| CivilComments | Acc. ↑ | $90.3_{\pm1.0}$ | $91.3_{\pm0.1}$ | $91.4_{\pm0.1}$ | $88.6_{\pm1.0}$ | $88.6_{\pm0.8}$ | $91.5_{\pm0.1}$ | $88.6_{\pm0.1}$ | $90.1_{\pm0.7}$ | $90.8_{\pm0.5}$ |
| | ECE ↓ | $10.4_{\pm0.4}$ | $4.8_{\pm0.2}$ | $7.8_{\pm1.7}$ | $11.4_{\pm0.5}$ | $2.4_{\pm0.4}$ | $4.2_{\pm0.1}$ | $11.1_{\pm0.1}$ | $\mathbf{2.1_{\pm0.8}}$ | $\underline{4.2_{\pm1.0}}$ |
| | CECE ↓ | $10.4_{\pm0.5}$ | $5.6_{\pm0.1}$ | $8.1_{\pm1.8}$ | $11.4_{\pm0.5}$ | $2.4_{\pm0.4}$ | $4.8_{\pm0.3}$ | $11.1_{\pm0.2}$ | $\mathbf{2.2_{\pm0.3}}$ | $\underline{4.6_{\pm0.8}}$ |
| | KSE ↓ | $10.4_{\pm0.4}$ | $6.7_{\pm0.1}$ | $7.7_{\pm1.7}$ | $5.8_{\pm0.1}$ | $2.9_{\pm0.4}$ | $4.3_{\pm0.1}$ | $11.0_{\pm0.1}$ | $\mathbf{3.3_{\pm0.8}}$ | $\underline{4.2_{\pm1.0}}$ |

Table 1: We report the OOD test scores (%) computed across 3 seeds, evaluated on both synthetic and wild benchmarks for *Peacock* and recent baselines. *Peacock* greatly improves calibration and maintains model accuracy. The best calibration scores in bold, second best are underlined.

## 5 EXPERIMENTS AND ANALYSIS

### 5.1 EXPERIMENT SETUP

**Evaluation Metrics** Following (Guo et al., 2017; Mukhoti et al., 2020; Neo et al., 2024), we use the Expected Calibration Error (ECE), Classwise Calibration Error (CECE) (Nixon et al., 2019) and Kolmogorov-Smirnov Error (KSE) (Gupta et al., 2021) for evaluation. For fair comparisons, we follow the evaluation protocols of other authors and compute calibration errors using 15 bins with the mean and standard deviation shown across seeds. Additional details of each metric are included in Appendix A.5.

**Datasets** We evaluate *Peacock* on a total of eight OOD image and text benchmarks. For synthetic datasets, we use CIFAR (Krishnan & Tickoo, 2020) and TinyImageNet (Deng et al., 2009) for training/validation and CIFAR-C/TinyImageNet-C (Hendrycks & Dietterich, 2019) for testing. For Wild datasets, we use Camelyon-17 (Bandi et al., 2019), iWildCam (Beery et al., 2020), FmoW (Christie et al., 2018), Amazon (Ni et al., 2019) and CivilComments(Borkan et al., 2019) from the Wilds benchmark (Koh et al., 2021). OOD data is never used for training or validating a model, only for testing.

**Baselines** We compare equal and importance weighted *Peacock* against an uncalibrated baseline (CE) and six components, specifically MaxEnt (Neo et al., 2024), AdaFocal (Ghosh et al., 2022), RankMixup (Noh et al., 2023), CPC (Cheng & Vasconcelos, 2022), Dual (Tao et al., 2023), ACLS (Park et al., 2023). For our analysis on image tasks, we use ResNet-18, ResNet-50 (He et al., 2016), SWINV2 (Liu et al., 2022b) and RoBERTa Liu et al. (2019b) for text tasks. We perform post-hoc processing with AdaTS (Joy et al., 2023) and compare different weighted formulations analyzing the overall contributions of each component used in *Peacock*. For additional details of each dataset task, hyper-parameters and illustrations of synthetic and wild OOD shifts, we refer readers to Appendix D.1.

| Dataset | ECE↓ | CE | MaxEnt | AdaFocal | RankMixup | CPC | Dual | ACLS | Peacock (Eq.) | Peacock (Impt.) |
|---|---|---|---|---|---|---|---|---|---|---|
| CIFAR10-C | Pre | $14.5_{\pm0.4}$ | $6.5_{\pm0.2}$ | $6.8_{\pm0.2}$ | $11.9_{\pm0.3}$ | $11.2_{\pm0.2}$ | $7.4_{\pm0.3}$ | $11.2_{\pm0.4}$ | $\underline{6.3}_{\pm0.1}$ | $\mathbf{6.2}_{\pm0.3}$ |
| | Post | $7.5_{\pm0.1}$ | $6.9_{\pm0.2}$ | $6.9_{\pm0.4}$ | $7.1_{\pm0.3}$ | $7.0_{\pm0.1}$ | $7.3_{\pm0.3}$ | $7.3_{\pm0.1}$ | $\mathbf{6.6}_{\pm0.1}$ | $6.9_{\pm0.3}$ |
| | Avg. | $11.0_{\pm0.3}$ | $6.7_{\pm0.2}$ | $6.9_{\pm0.3}$ | $9.5_{\pm0.3}$ | $9.1_{\pm0.2}$ | $7.4_{\pm0.3}$ | $9.3_{\pm0.3}$ | $\mathbf{6.4}_{\pm0.1}$ | $\underline{6.6}_{\pm0.3}$ |
| CIFAR100-C | Pre | $10.6_{\pm0.1}$ | $11.6_{\pm0.6}$ | $13.6_{\pm0.1}$ | $11.0_{\pm1.4}$ | $13.2_{\pm0.5}$ | $15.7_{\pm0.1}$ | $12.2_{\pm0.2}$ | $\underline{9.6}_{\pm0.2}$ | $\mathbf{9.3}_{\pm0.1}$ |
| | Post | $8.1_{\pm0.4}$ | $\mathbf{7.2}_{\pm0.1}$ | $\underline{7.5}_{\pm0.1}$ | $8.0_{\pm0.1}$ | $10.2_{\pm0.2}$ | $8.4_{\pm0.2}$ | $8.1_{\pm0.4}$ | $8.7_{\pm0.2}$ | $8.9_{\pm0.2}$ |
| | Avg. | $9.4_{\pm0.3}$ | $9.4_{\pm0.4}$ | $10.6_{\pm0.1}$ | $9.5_{\pm1.0}$ | $11.7_{\pm0.3}$ | $12.1_{\pm0.2}$ | $10.2_{\pm0.3}$ | $\underline{9.2}_{\pm0.2}$ | $\mathbf{9.1}_{\pm0.2}$ |
| TinyImageNet-C | Pre | $15.7_{\pm0.5}$ | $12.8_{\pm0.1}$ | $13.8_{\pm0.4}$ | $20.2_{\pm0.1}$ | $16.0_{\pm0.5}$ | $19.2_{\pm0.4}$ | $19.8_{\pm0.2}$ | $\underline{10.6}_{\pm0.2}$ | $\mathbf{10.4}_{\pm0.2}$ |
| | Post | $25.4_{\pm0.3}$ | $20.2_{\pm0.2}$ | $26.1_{\pm0.4}$ | $24.3_{\pm0.3}$ | $20.7_{\pm0.4}$ | $24.3_{\pm0.3}$ | $24.4_{\pm0.4}$ | $\mathbf{11.9}_{\pm0.2}$ | $\underline{12.6}_{\pm0.2}$ |
| | Avg. | $20.5_{\pm0.4}$ | $16.5_{\pm0.2}$ | $20.0_{\pm0.4}$ | $22.3_{\pm0.3}$ | $18.4_{\pm0.5}$ | $21.8_{\pm0.4}$ | $22.1_{\pm0.3}$ | $\mathbf{11.2}_{\pm0.4}$ | $\underline{11.5}_{\pm0.4}$ |
| Camelyon17 | Pre | $15.5_{\pm1.1}$ | $12.4_{\pm0.1}$ | $13.8_{\pm0.4}$ | $20.2_{\pm0.1}$ | $16.0_{\pm0.5}$ | $19.2_{\pm0.4}$ | $19.8_{\pm0.2}$ | $11.7_{\pm0.7}$ | $\mathbf{9.87}_{\pm1.8}$ |
| | Post | $33.1_{\pm0.4}$ | $11.2_{\pm1.1}$ | $15.4_{\pm1.5}$ | $14.9_{\pm1.4}$ | $12.2_{\pm1.4}$ | $18.1_{\pm0.3}$ | $11.6_{\pm0.7}$ | $\mathbf{9.87}_{\pm3.0}$ | $12.2_{\pm1.7}$ |
| | Avg. | $24.3_{\pm0.8}$ | $11.8_{\pm0.6}$ | $14.6_{\pm1.0}$ | $17.6_{\pm0.8}$ | $14.1_{\pm0.9}$ | $18.7_{\pm0.4}$ | $15.7_{\pm0.4}$ | $\mathbf{10.8}_{\pm1.5}$ | $\underline{11.0}_{\pm1.8}$ |
| iWildCam | Pre | $30.6_{\pm0.8}$ | $21.0_{\pm3.2}$ | $23.0_{\pm0.5}$ | $25.5_{\pm0.7}$ | $20.3_{\pm1.1}$ | $13.0_{\pm2.5}$ | $20.6_{\pm1.8}$ | $\underline{12.0}_{\pm0.1}$ | $\mathbf{11.7}_{\pm2.3}$ |
| | Post | $8.0_{\pm2.2}$ | $8.6_{\pm0.6}$ | $7.9_{\pm1.3}$ | $9.5_{\pm2.5}$ | $11.7_{\pm1.9}$ | $8.0_{\pm1.0}$ | $\underline{7.9}_{\pm0.4}$ | $8.9_{\pm0.6}$ | $\mathbf{6.7}_{\pm0.5}$ |
| | Avg. | $19.3_{\pm1.5}$ | $14.8_{\pm2.4}$ | $15.5_{\pm1.1}$ | $17.5_{\pm1.6}$ | $16.0_{\pm1.6}$ | $10.5_{\pm1.6}$ | $14.3_{\pm1.2}$ | $\underline{10.5}_{\pm0.4}$ | $\mathbf{9.2}_{\pm1.4}$ |
| FmoW | Pre | $39.8_{\pm0.2}$ | $20.0_{\pm9.9}$ | $20.9_{\pm8.6}$ | $41.7_{\pm0.1}$ | $22.4_{\pm0.9}$ | $10.7_{\pm0.1}$ | $21.7_{\pm0.2}$ | $\underline{10.6}_{\pm0.4}$ | $\mathbf{10.5}_{\pm0.3}$ |
| | Post | $25.6_{\pm0.6}$ | $\mathbf{4.9}_{\pm0.9}$ | $6.2_{\pm0.5}$ | $7.9_{\pm0.3}$ | $7.6_{\pm0.9}$ | $5.6_{\pm0.9}$ | $6.1_{\pm0.5}$ | $\underline{5.3}_{\pm0.8}$ | $5.9_{\pm0.5}$ |
| | Avg. | $32.7_{\pm0.5}$ | $12.5_{\pm5.5}$ | $13.6_{\pm4.7}$ | $24.8_{\pm0.5}$ | $15.0_{\pm0.8}$ | $8.2_{\pm0.2}$ | $13.9_{\pm0.2}$ | $\mathbf{7.9}_{\pm0.4}$ | $\underline{8.2}_{\pm0.8}$ |

Table 2: ECE (%) scores before and after AdaTS (Joy et al., 2023) for the different OOD datasets. *Peacock* delivers the best overall calibration performance, despite using temperatures obtained ID.

## 5.2 COMPARISONS TO PUBLISHED BASELINES

**In-Distribution Performance** Fig. 1 and Table 5 showcase the ID results, demonstrating *Peacock*'s consistently strong calibration performance across datasets. Following Eq. (12), *Peacock*'s ECE is significantly lower than the empirical average $\overline{\text{ECE}}$ of all methods. While individual algorithms may vary in performance across datasets, we demonstrate that combining them through *Peacock* consistently improves calibration - with no significant loss in accuracy. Additional ID results and discussions are included in Appendix D.2.

**Out-of-Distribution Performance** Similar to the ID results, each individual algorithm's independent performance varies across different datasets. Table 1 shows that by combining algorithms through equal importance *Peacock*, consistent improvements in OOD performance can be achieved on both synthetic and real-world image and text datasets. Our findings further demonstrate that the equal importance formulation of *Peacock* also adheres to the theoretical upper bound in Eq. (12) for OOD performance. Finally, *Peacock* can be further enhanced using our weighted importance formulation, achieving improved results and good generalization properties across all datasets.

**Training Time per Epoch** Fig. 4 shows the average wall-clock time per epoch (forward pass, loss-calculation and back-propagation) for each method trained on CIFAR10. We report the wall-clock time in seconds on a NVIDIA GeForce RTX 2070 GPU with i7-10700 CPU. In general, each algorithm has similar speeds, with RankMixup taking the longest since another forward pass is required to obtain the logits of interpolated samples. On the other hand, *Peacock* is optimized (see Appendix B.2) to remain competitive with other baselines, despite combining multiple algorithms together.

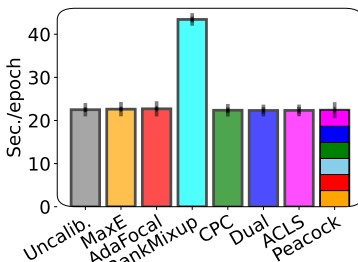

Figure 4: Wall-clock time for each method on CIFAR10. *Peacock* is as fast as each of its components.

## 5.3 POST-HOC PROCESSING

For post-hoc calibration, we apply AdaTS (Joy et al., 2023) to each method. The samplewise temperatures are obtained from an ID validation set and applied to the OOD test sets. Table 2 presents the ECE scores of each algorithm before and after applying AdaTS for all six OOD image datasets. Our findings demonstrate that *Peacock* delivers the best overall calibration performance, both before and after temperature scaling. In cases where *Peacock* does not deliver the best calibration, we can see that its performance is relatively close to the best score. While AdaTS generally improves OOD ECE, applying it to already well-calibrated models can sometimes lead to degraded performance. For instance, MaxEnt Loss achieves the best OOD calibration with 4.9% on FmoW without AdaTS, but applying it subsequently would cause the ECE to worsen to 12.5%. This discrepancy can be

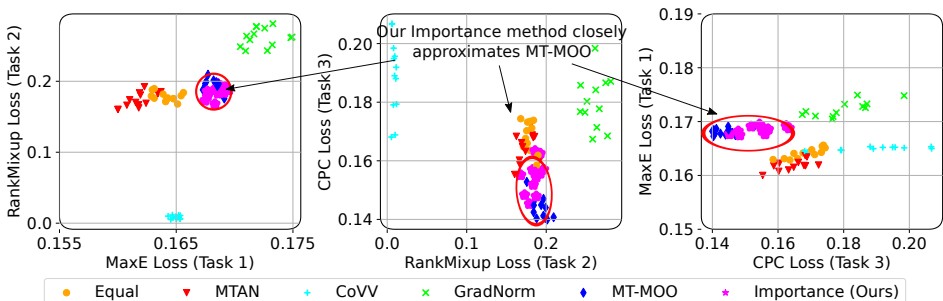

Figure 5: Solution given by different MOO algorithms for each of the auxiliary test losses. Our importance formulation effectively balances trade-offs between loss terms. Bottom-left is better.

| Algorithm | Acc (%) | ECE (%) | Speed (Sec) | w1 | w2 | w3 | $\sum_t^A w_t = 1$ |
|---|---|---|---|---|---|---|---|
| Equal-Importance | $51.8_{\pm 0.1}$ | $9.6_{\pm 0.2}$ | $50.1_{\pm 0.2}$ | 0.33 | 0.33 | 0.33 | Yes |
| MTAN (Liu et al., 2019a) | $50.8_{\pm 0.1}$ | $10.2_{\pm 0.4}$ | $50.7_{\pm 0.2}$ | 0.33 | 0.33 | 0.33 | Yes |
| CoVV (Groenendijk et al., 2021) | $52.6_{\pm 0.1}$ | $12.0_{\pm 0.4}$ | $50.6_{\pm 0.2}$ | 0.01 | 0.52 | 0.47 | Yes |
| GradNorm (Chen et al., 2018) | $48.9_{\pm 0.6}$ | $9.3_{\pm 0.7}$ | $85.8_{\pm 0.7}$ | 2.99 | 0.00 | 0.00 | No |
| MT-MOO (Sener & Koltun, 2018) | $51.9_{\pm 0.5}$ | $9.3_{\pm 0.5}$ | $67.5_{\pm 0.5}$ | 0.36 | 0.47 | 0.16 | Yes |
| Weighted-Importance (Ours) | $52.6_{\pm 0.1}$ | $9.3_{\pm 0.3}$ | $50.5_{\pm 0.2}$ | 0.00 | 0.51 | 0.49 | Yes |

Table 3: Comparisons of different multi-objective optimization methods for *Peacock*. Our weighted importance formulation is fast and effective.

attributed to the disconnect between the training, validation sets and test set, explaining the higher calibration errors after temperature scaling (Ovadia et al., 2019). However, by combining multiple calibration algorithms together, *Peacock* displays the best generalization behavior even when using temperatures obtained ID. As AdaTS is designed solely for image tasks, we apply vanilla TS for Amazon and CivilComments indicating their results and ideal temperatures obtained from grid-search in Table 7. We further highlight that temperature scaling only manipulates the predicted confidences and does not affect recognition accuracy.

## 5.4 Weighted Peacock Performance and Analysis

We compare various weighted objective optimization methods for *Peacock*. Namely, using equal weights, MTAN (Liu et al., 2019a), GradNorm (Chen et al., 2018), CoVV (Groenendijk et al., 2021), MT-MOO (Sener & Koltun, 2018) and our proposed weighted importance variant, on CIFAR100/100-C using ResNet-18. All MOO methods are initialized with uniform weights, with the final test accuracy, ECE and weights shown upon convergence with the average training wall clock time per epoch in Table 3. Our results indicate that while each algorithm yields distinct solutions/weights for each loss term, they achieve comparable accuracies and calibration errors. GradNorm and MT-MOO have longer training times (about 65% and 35% respectively) since they require the recomputation of $\nabla_\theta \mathcal{L}_t(\theta)$ for each loss term. Conversely, MTAN and CoVV offers faster performance, but has higher ECE. Fig. 5 further demonstrates that the auxiliary test losses for each MOO method yield similar solutions, with MT-MOO achieving the optimal solution. Our method closely approximates MT-MOO but is faster and more efficient. We further discuss contributions of each calibration loss term in Appendix D.3 and the limitations of our work in Appendix E.

## 6 Conclusions

We present *Peacock*, a unified framework for neural network calibration. By formulating unification as a multi-objective optimization problem, we demonstrate that combining calibration components improves performance on both ID and OOD tasks. Our proposed weighted importance form of *Peacock* is fast and effective in delivering good Pareto Optimal performance. Despite incorporating multiple algorithms, *Peacock*'s complements post-hoc processing and remains fast in terms of computational speed. Our method shows clear performance gains with RankMixup and MaxEnt loss offering the most improvements.

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

## A    Calibration Algorithms and Metrics

In this section, we further discuss in detail the various families of approaches commonly used to improve and measure neural network calibration.

### A.1    Entropy-based Methods

Entropy-based methods have played an important role in calibrating deep neural networks, as maximizing the entropy helps penalize overconfident predictions (Pereyra et al., 2017; Mukhoti et al., 2020; Neo et al., 2024). As mentioned in the main text, naively penalizing all predictions can cause underconfident predictions. While various works have proposed different approaches in controlling the entropy term, the Focal Loss (Lin et al., 2017; Mukhoti et al., 2020; Ghosh et al., 2022) and it's variants offer adaptive/automated mechanisms in obtaining suitable values of $\gamma$ for each sample.

While these automated mechanisms tend to help with ID calibration, many works fail to acknowledge the importance of OOD calibration since the parameters obtained during training/validation may not work during testing (Ovadia et al., 2019). As a work-around, we find that entropy-based methods can be extended to include OOD Maximum Entropy constraints (Jaynes, 1957; Neo et al., 2024) or Dual logit manipulation (Tao et al., 2023), showcasing the versatility of entropy-based methods. Since these methods all share the form of the Focal loss, we can easily pair all of them together into a single step.

### A.2    Regularizers

Mixup is an effective regularization technique that augments (Zhang et al., 2018) both input features and labels. Mixup works particularly well on both wider and deeper networks (Zhang et al., 2022) and can be particularly useful in improving network calibration (Thulasidasan et al., 2019; Chidambaram & Ge, 2024). As an extension to vanilla Mixup, RankMixup (Noh et al., 2023) can be used to ensure that the augmented samples have lower confidences than the original samples.

### A.3    Margin-based Methods

Margin-based methods tend to restrict model confidences by a constant margin/factor. For example, label smoothing (LS) (Müller et al., 2019) softens the targets using a constant factor $\epsilon$. Mathematically, the smoothed label $s_i$ is acquired after uniformly adjusting the target $s_i = (1 - \epsilon)y_k + \frac{\epsilon}{K}$, which is then used to train the network. Although vanilla LS can be used to improve miscalibration, imposing a constant smoothing factor for all training labels can lead to under-confident predictions. Furthermore, searching for a suitable $\epsilon$ is computationally expensive as it requires a grid-search across multiple models during the training phase.

Instead of implementing a fixed constant, several works have been proposed to adaptively or conditionally approximate the label smoothing function during training. For example, MDCA (Hebbalaguppe et al., 2022) utilizes a regularization term, which enforces predicted confidences to be as close to the average accuracy as possible. This can lead to a parabolic smoothing function (Park et al., 2023), that is adaptively dependent on the predicted confidences. Which can be problematic, since both high and low confidence predictions are weakly penalized. Another approach would be to only conditionally smooth predictions based on a margin. For instance, MBLS (Liu et al., 2022a) and CALS-ALM (Liu et al., 2023a) propose to restrict output logits by a user defined margin, but can be sensitive to hyper parameter settings. CRL (Moon et al., 2020) ordinarily ranks predictions based on the number of times each sample is predicted correctly, however it requires a buffer to store the correctness history. Which can be empty during the earlier stages of training and idle during later phases when the model's accuracy is high.

By adopting a smoothing and indicator function, the Adaptive Conditional Label Smoothing (ACLS) (Park et al., 2023) method seeks to combine the benefits of both adaptive and conditional methods without the use of an additional correctness history.

## A.4 POST-HOC PROCESSING

The fundamental idea behind post-hoc processing methods is to obtain a mapping function/temperature that modifies the model's logits thus changing it's predicted confidence. The most popular post-processing step is the vanilla temperature scaling (TS) (Platt & Karampatziakis, 2007), which manipulates the model's confidences without changing the final class label predictions. For example, a value of $T < 1$ leads to a lower entropy or "peaky" distributions and a value of $T > 1$ gives higher entropy or "flatter" predictions.

The typical approach in obtaining the temperature parameter, is to minimize the *average* calibration error or NLL over a seperate valdation set. While vanilla TS has been found to be effective in reducing network over-confidence (Guo et al., 2017), it generally reduces the confidence of every sample - even when predictions are correct. Other forms of post-hoc processing include calibration using, model ensembles (Zhang et al., 2020), splines (Gupta et al., 2021) and distribution matching (Kuleshov & Deshpande, 2022; Tomani et al., 2023). For our post-processing step, we use AdaTS since it is the SOTA method for post-processing methods and adaptively chooses a samplewise temperature for scaling model predictions.

## A.5 CALIBRATION METRICS

**Expected Calibration Error (ECE):** The ECE is the most widely used metric in the literature and directly tied to the definition of calibration (Guo et al., 2017; Tomani et al., 2021). By splitting the predicted confidences in $B$ evenly separated bins, each containing $n_b$ samples. The ECE is then simply a scalar measuring the weighted errors between the *acc* and *conf* of each bin (Naeini et al., 2015): ECE $= \sum_{b=1}^{B} \frac{n_b}{N} |acc(b) - conf(b)|$. Despite the ECE's popularity, many recent works have pointed out the limitations of the ECE, such as bin size sensitivity and it's lack of consideration for classwise calibration. For a fair and thorough analysis, we introduce other calibration metrics that cover the weaknesses of the ECE.

**Classwise ECE (CECE):** As most calibration metrics typically only considers the max confidence probabilities, the CECE considers the macro-averaged ECE of all $K$ classes. Predictions are binned individually for each respective class and the calibration error is measured for each class level bin (Nixon et al., 2019). CECE $= \frac{1}{K} \sum_{b=1}^{B} \sum_{k=1}^{K} \frac{n_{b,k}}{N} |acc(b,k) - conf(b,k)|$.

**Overconfidence Error (OE):** For safety-critical applications, overconfident mispredictions are potentially hazardous. The OE penalizes overconfident bins that have higher confidences than accuracy (Thulasidasan et al., 2019): OE $= \sum_{b=1}^{B} \frac{n_b}{N} \left[ conf(b) \times \max(conf(b) - acc(b), 0) \right]$.

**Kolmogorov-Smirnov Error (KSE):** As many calibration metrics are often sensitive to the number of $B$ bins used during the partitioning of empirical distributions. The KSE(Gupta et al., 2021) is a bin-free alternative that numerically approximates the differences between two empirical cumulative distributions. The KSE for top-1 classification is given as the following integral, with $z_k$ denoting the predicted probabilities: KSE $= \int_0^1 |P(k|z_k) - z_k| P(z_k) dz_k$.

**Adaptive ECE (AdaECE:** as the ECE is known to be biased towards higher confidence bins, the AdaECE (Nguyen & O'Connor, 2015) is proposed to adaptively/evenly measure samples across bins: AdaECE $= \sum_{b=1}^{B} \frac{n_b}{N} |acc(b) - conf(b)|$ s.t. $\forall b, i \cdot |B_b| = |B_i|$.

**Negative Log-likelihood (NLL):** Commonly referred to as cross entropy in deep learning. The NLL (Hastie et al., 2001) measures the alignment between a model's confidence $P_i(y_k|x)$ and targets $y_k$: NLL $= -\frac{1}{N} \sum_{i=1}^{N} \sum_{k=1}^{K} y_k \log P_i(y_k|x)$.

---

**Algorithm 1:** Peacock - Unified Multi-Objective Optimization Calibration Framework

---

**Data:** Given training and validation set $D_{\text{train}} = (x_i, y_i)_{i=1}^N$, $D_{\text{val}} = (x_v, y_v)_{v=1}^V$

1: Initialize neural network parameters $\boldsymbol{\theta}$, learning rate schedule $\eta$ and uniformly distributed weights $w_t = \frac{1}{A}$
2: Compute the global and local expectations for the mean and variance constraints $\mu, \sigma^2$
3: $\quad\hookrightarrow \mathbb{E}[\mathcal{Y}] = \mu$ and $\mathbb{E}[\mathcal{Y}^2] = \sigma^2$
4: Solve numerically for $\lambda_\mu \leftarrow \texttt{NewtonRaphson()}$       // MaxEnt loss root-finder
5: **for** $e \in epochs$ **do**
6:    **for** $i \in B$ **do**       // Sample mini-batch of size $B$
7:       Perform FastMixup on images: $\tilde{x} = \beta x_i + (1 - \beta) x_j$       // RankMixup
8:       Perform FastMixup on labels: $\tilde{y} = \beta y_i + (1 - \beta) y_j$       // RankMixup
9:       Compute 1v1 loss: $\mathcal{L}_{\text{CPC}}^{\text{1v1}} = -\frac{1}{(C-1)} \sum_{j \neq y} \log \frac{P_i y}{P_i y + P_i j}$       // CPC loss
10:       **if** $\gamma_{t,b} \geq 0$ **then**
11:         $\mathcal{L}_{\text{AdaFocal}}^{\text{Dual}} = -\sum_k (1 - P_i + P_j)^{\gamma_{t,b}} \log P_i$       // Dual AdaFocal loss
12:       **else if** $\gamma_{t,b} < 0$ **then**
13:         $\mathcal{L}_{\text{AdaFocal}}^{\text{Dual}} = -\sum_k (1 + P_i + P_j)^{|\gamma_{t,b}|} \log P_i$       // Inverse Dual AdaFocal loss
14:       Compute MaxE loss $\mathcal{L}_{\text{ME}} = \lambda_\mu (\sum_k f(\mathcal{Y}) P_i(y_k|x) - \mu_G + \sum_k f(\mathcal{Y}) P_i(y_k|x) - \mu_{Lk})$       // MaxEnt loss
15:       Compute MRL loss $\mathcal{L}_{\text{MRL}} = \max(0, \max_k \tilde{P} - \max_k P + m_{\text{MRL}})$       // RankMixup
16:       **if** $j = \hat{y}$ **then**
17:         $\mathcal{L}_{\text{ACLS}} = \lambda_1 \max(0, g_j^\theta(x) - \min_k(g_k^\theta(x)) - m_{\text{ACLS}})^2$       // ACLS regularizer
18:       **else if** $j \neq \hat{y}$ **then**
19:         $\mathcal{L}_{\text{ACLS}} = \lambda_2 \max(0, g_{\hat{y}}^\theta(x) - g_j^\theta(x) - m_{\text{ACLS}})^2$       // ACLS regularizer
20:       $w_t = \texttt{ImportancePeacock}(\mathcal{L}_t(\boldsymbol{\theta}))$       // Compute importance loss weights
21:
22:       Compute Peacock:
23:         $\hookrightarrow \quad \mathcal{L}_{\text{Peacock}} = \mathcal{L}_{\text{AdaFocal}}^{\text{Dual}} + w_1 \mathcal{L}_{\text{constraints}}^{\text{ME}} + w_2 \mathcal{L}_{\text{CPC}}^{\text{1v1}} + w_3 \mathcal{L}_{\text{MRL}}$
24:       $\boldsymbol{\theta}_{\text{new}} \leftarrow \boldsymbol{\theta}_{\text{old}} - \eta \nabla_{\boldsymbol{\theta}} \mathcal{L}_{\text{Peacock}}$       // Update parameters $\boldsymbol{\theta}$ by gradient descent
25: **return** $\boldsymbol{\theta}$
26:
27: Apply temperature scaling: $\boldsymbol{\theta}_{\text{AdaTS}} \leftarrow \texttt{AdaptiveTS}(D_{val}, \boldsymbol{\theta})$       // AdaTS
28: **Function** $\texttt{NewtonRaphson()}$:
29:    $\delta = 1\text{e-}15$       // A small tolerance or stopping condition
30:    **while** $g(\lambda) > \delta$ **do**
31:       $\lambda_{n+1} = \lambda_n - \frac{g(\lambda)}{g'(\lambda)}$       // Update Lagrange Multipliers $\lambda_n$
32:    **return** $\lambda_n$
33: **Function** $\texttt{ImportancePeacock()}$:
34:    $\min_{w_t} \left\{ \left\| \sum_{t=1}^A w_t \sqrt{\frac{\Delta_{\boldsymbol{\theta}} \mathcal{L}_t(\boldsymbol{\theta})}{\eta}} \right\|_2^2 \Big| \sum_{t=1}^A w_t = 1, w_t \geq 0 \quad \forall t \right\}$
35:    **return** $w_t$
36: **Function** $\texttt{AdaptiveTS}(D_{val}, \boldsymbol{\theta})$:
37:    Initialize VAE and MLP parameters $\boldsymbol{Q}, \boldsymbol{\phi}$
38:    **while** $t < steps$ **do**
39:       **for** $v \in B$ **do**       // Sample mini-batch of size $B$
40:         $\nabla_{VAE} \leftarrow \nabla \mathbb{ELBO}[\Phi(x)]$
41:         $\tilde{q} = \{\log P(z|y) | \forall y\} \quad z \sim Q_\phi(z|x)$
42:         $\nabla_T \leftarrow \log(\text{softmax}(g_\theta/T))$
43:         $(\boldsymbol{\theta}, \phi)_{t+1} \leftarrow (\boldsymbol{\theta}, \phi)_t - \alpha_{\text{lr}}(\nabla_{VAE} + \nabla_T)$
44:    **return** $\boldsymbol{\theta}_{\text{AdaTS}}$

---

# B   IMPLEMENTATION DETAILS FOR PEACOCK

## B.1   ALGORITHM DETAILS AND HYPERPARAMETERS

For our implementation of *Peacock*, we first select the mean constraint for of MaxEnt loss as our starting algorithm and compute Lagrange multipliers $\lambda_n$ using the Newton Raphson method. This step is performed in $\mathcal{O}(n)$ time using the helper function $g(\lambda)$ and its derivative $g'(\lambda)$ before model training begins.

For each iteration, the pairwise 1v1 constraints of CPC loss are first computed before incorporating the adaptive $\gamma$ selection mechanism of AdaFocal loss. This step also includes the second highest confidence $P_i(y_j|x)$ from Dual Focal loss to AdaFocal loss. This way we can reduce compute overhead by combining both calibration methods into a single step: $\mathcal{L}_{\text{AdaFocal}}^{\text{Dual}} = \mathcal{L}_{\text{AdaFocal}} + \mathcal{L}_{\text{Dual}}$.

Next, RankMixup is performed for every sampled input image and label, with the MRL loss computed using the coefficients $\alpha$ and $m$. For texts datasets, we perform RankMixup at the feature level. Although vanilla Mixup has been found to hurt ID calibration performance (Wang et al., 2023), our findings suggest that by combining RankMixup with other algorithms good balance between ID and OOD calibration can be achieved. In our experience, we find that a large ACLS margin, can lead

| Hyperparameters | Values |
|---|---|
| Learning rate $\eta$ | 0.1 |
| Batch size | 512 or 256 |
| Optimizer | SGD or Adam |
| Scheduler | Cosine Annealing or Fixed |
| Epochs | 200 or 50 |
| Margin $m_{ACLS}$ | 6.0 |
| Mixup $\alpha$ | 1.0 |
| Mixup margin $m_{MRL}$ | 2.0 |
| $\gamma$ starting | 1.0 |
| $\gamma$ max | 20.0 |
| $\gamma$ min | -2.0 |
| No. of bins $B$ | 15.0 |
| Learning rate for attention block | 3e-4 |

Table 4: Hyperparameters used for optimizing *Peacock*

to numerical instability when the number of classes is large, thus we fixed $m_{ACLS} = 6.0$. For completeness, we include the ACLS step in Algorithm 1, however in our ablation study we show that ACLS does not improve overall calibration performance and is not included during optimization or our final proposed version of *Peacock*. Next, an optional post-processing step using AdaTS is performed by learning the adaptive temperature on a seperate validation set. Finally, for the importance weighted form of *Peacock*, we randomly initialize a self-attention block and optimize it with Eq. (16) with Adam optimizer and a learning rate of 3e-4 to learn a set of importance weights for each loss term.

**Hyperparameters** In general, we try to keep the default settings of each algorithm. However, when trying to combine multiple of these components, it may become inevitable for some tuning to be performed. Indeed, performing a grid-search would be the best way to obtain the optimal hyperparameters. However, as discussed in our *Limitations*, the number of parameters scale exponentially with the number of calibration components selected for optimization. This can be easily become very compute intensive and would not be the focus of our work.

### B.2 ACCELERATING RANKMIXUP

Fig. 6 illustrates the comparisons between the original RankMixup method and the optimized version proposed in our paper. RankMixup, in its original form, requires two forward passes during training: one for a full minibatch (e.g., 512) of original images and another full minibatch of mixed images. This process can be computationally expensive, especially for large datasets or complex

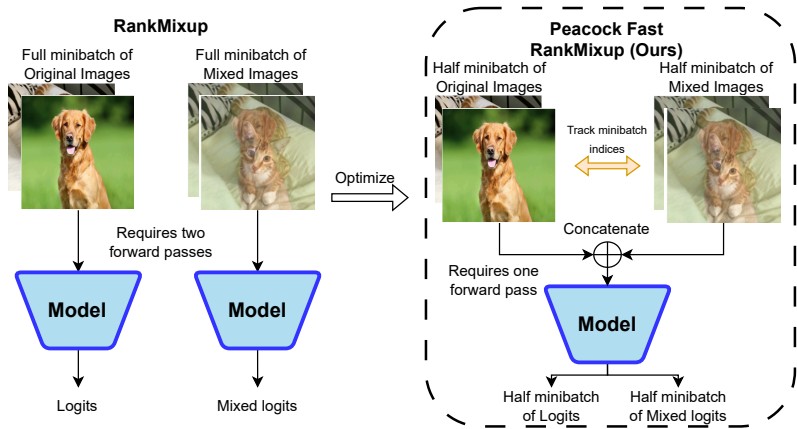

Figure 6: During training, RankMixup requires two forward passes: one for original images and one for mixed images, in order to compute $\mathcal{L}_{MRL}$. We optimize RankMixup by mixing images and labels batchwise, resulting in a 2x speed up during training.

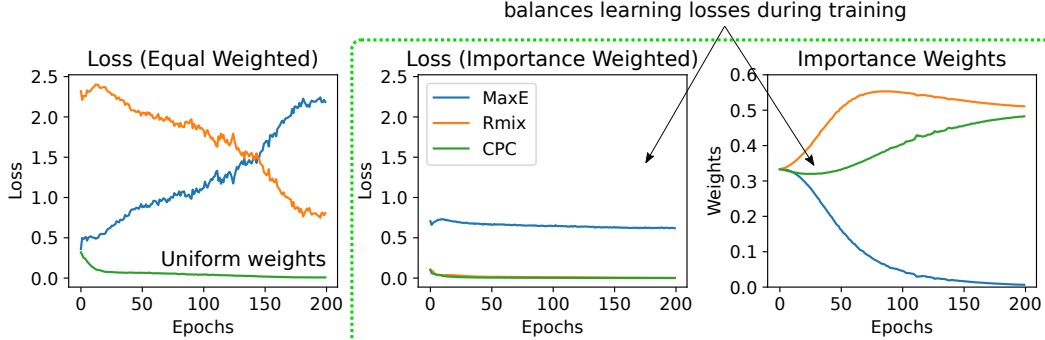

Figure 7: When using equal weights (left), the model optimizes loss terms equally. Our direction-weighted self-attention block, on the other hand, learns to dynamically adjust the importance of each loss term during training, enabling a more balanced optimization of the overall objective. All weights are initialized uniformly with plots smoothed for readability.

models. As a workaround, we propose an optimized variant of FastRankMixup, which addresses this limitation by dividing a full batch of images into two halves: containing a minibatch of half original and half mixed images (e.g., $512 \div 2 = 256$).

This way, we only require a single forward pass instead of the two forward passes, delivering a 2x speedup during training compared to the original RankMixup implementation. This improvement in training efficiency can be particularly beneficial for large-scale training tasks, where computational resources are often constrained. A caveat to this method is that the minimum batchsize required will always be two, as at least two samples are nee                                    formed.

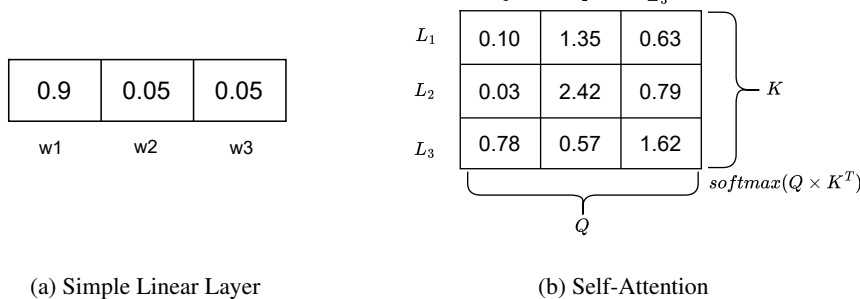

(a) Simple Linear Layer                (b) Self-Attention

Figure 8: Comparisons between a simple linear layer versus self-attention for *Peacock*. Self-attention is better suited for capturing the complex relationships between loss terms.

### B.3 LEARNING DYNAMICS OF DIRECTION WEIGHTED SELF-ATTENTION BLOCK

The importance-weighted formulation of *Peacock* utilizes a novel direction weighted self-attention block. This subsection discusses the learning dynamics and certain key considerations of the self-attention block. Fig. 7 illustrates the differences between the learning dynamics of the equal and importance weighted *Peacock* on CIFAR100. By assigning equal weights to each loss term, the model regards all auxiliary losses equally. Conversely, our proposed direction weighted self-attention block outputs importance weights at every timestep, using Eq. (16). This leads to an overall balanced and more stable learning process during optimization. Note that all loss terms are normalized before being passed into the self-attention block during training. Additionally, the direction weighted self-attention block provides certain key benefits:

- **Softmax of Self-Attention:** The softmax function of the self-attention block implicitly enforces KKT conditions, simplifying the optimization process.
- **Better learns relationships across losses:** The design of self-attention enables better learning of inter-dependencies among loss terms, compared to a linear layer (see Fig. 8).

## C    Proofs

### C.1    Temperature-scaled bounds

Consider a temperature/mapping function $T$ which scales the output logits/hypothesis $h^{\boldsymbol{\theta}}$ of a model. Then the average of each temperature scaled hypothesis is given as:

$$\overline{\text{ECE}}^2(\mathcal{T}(h^{\boldsymbol{\theta}})) = \frac{1}{A}\sum_{t=1}^{A}\text{ECE}^2(\mathcal{T}(h_t^{\boldsymbol{\theta}})) = \frac{\text{ECE}(\mathcal{T}(h_1^{\boldsymbol{\theta}}))^2 + \text{ECE}(\mathcal{T}(h_2^{\boldsymbol{\theta}}))^2 + ... + \text{ECE}(\mathcal{T}(h_t^{\boldsymbol{\theta}}))^2}{A} \tag{17}$$

Considering equal contributions of each individual temperature scaled hypothesis, the temperature scaled multi-objective learner $\mathcal{T}(H^{\boldsymbol{\theta}})$ has the expected squared ECE:

$$\text{ECE}^2(\mathcal{T}(H^{\boldsymbol{\theta}})) = \mathbb{E}\left[\left(\frac{1}{A}\sum_{t=1}^{A}\text{ECE}(\mathcal{T}(h_t^{\boldsymbol{\theta}}))\right)^2\right] = \int\left(\frac{1}{A}\sum_{t=1}^{A}\text{ECE}(\mathcal{T}(h_t^{\boldsymbol{\theta}}))\right)^2 p(x)dx \tag{18}$$

which follows the same bounds as previously defined in the main paper.

$$\text{ECE}^2(\mathcal{T}(H^{\boldsymbol{\theta}})) \leq \overline{\text{ECE}}^2(\mathcal{T}(h^{\boldsymbol{\theta}})) \tag{19}$$

Empirically, Table 2 demonstrates that if the same mapping function or temperature $\mathcal{T}$ is applied to each hypothesis (e.g., AdaTS), then the average of the scaled combined learner will also obey the upper bound of the above inequality.

### C.2    Estimating the Gradient

Recall in Section 4.2 of our main paper, the direct computation of $\nabla_{\boldsymbol{\theta}}\mathcal{L}_t(\boldsymbol{\theta})$ requires the use of retaining the computational graph[4] after the backward pass, which can be compute intensive and significantly slows down training time. In this section, we demonstrate that decrease rate estimates for each loss term can act as alternatives to direct gradient recomputation. By simply storing the previous loss value computed (single step look-back), we can avoid graph retention during the optimization for $w_t$. Using a simple example, we also show that our decrease rate estimates are closely related to the solutions obtained using gradient descent. For simplicity, we denote the partial derivatives as $\nabla_{\boldsymbol{\theta}}\mathcal{L}_t(\boldsymbol{\theta}) = \frac{\partial \mathcal{L}_t(\boldsymbol{\theta})}{\partial(\boldsymbol{\theta})}$ and the difference between old and new parameters as $\Delta_{\boldsymbol{\theta}}\mathcal{L}_t(\boldsymbol{\theta})$.

---

[4] For more details, see Pytorch autograd framework: *https://pytorch.org/docs/stable/autograd.html*

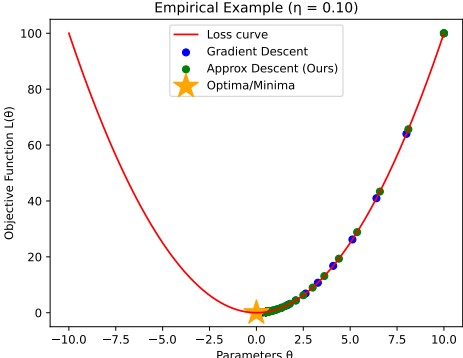
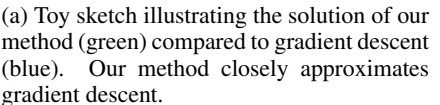
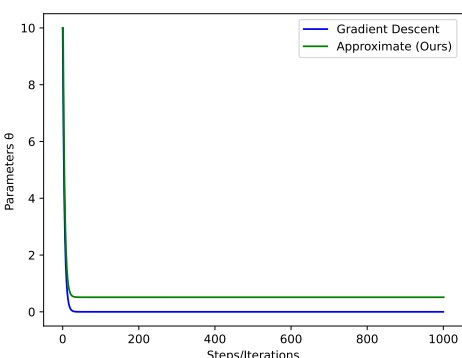

(a) Toy sketch illustrating the solution of our method (green) compared to gradient descent (blue). Our method closely approximates gradient descent.

(b) We compare the solutions given by our method and gradient descent, for $\eta = 0.1$, our solution is close to the solution by gradient descent.

Consider the following task of approximating $\nabla_{\boldsymbol{\theta}}\mathcal{L}_t(\boldsymbol{\theta})$. By using the first order form of Taylor's Theorem, the loss gradients can be rewritten as the following equation:

$$\nabla_{\boldsymbol{\theta}}\mathcal{L}_t(\boldsymbol{\theta}) = \frac{\mathcal{L}_t(\boldsymbol{\theta}_{\mathrm{new}}) - \mathcal{L}_t(\boldsymbol{\theta}_{\mathrm{old}})}{\Delta\boldsymbol{\theta}} + \epsilon(\boldsymbol{\theta}) = \frac{\Delta_{\boldsymbol{\theta}}\mathcal{L}_t(\boldsymbol{\theta})}{\Delta\boldsymbol{\theta}} + \epsilon(\boldsymbol{\theta}) \tag{20}$$

where $\Delta_{\boldsymbol{\theta}}\mathcal{L}_t(\boldsymbol{\theta})$ is the rate of change for each loss term with respect to the change of model parameters $\boldsymbol{\theta}_{\mathrm{new}}$ and $\boldsymbol{\theta}_{\mathrm{old}}$, paired by a small error term $\epsilon(\boldsymbol{\theta})$. From the gradient descent update rule, the change in model parameters is given by:

$$\boldsymbol{\theta}_{\mathrm{new}} = \boldsymbol{\theta}_{\mathrm{old}} - \eta\nabla_{\boldsymbol{\theta}}\mathcal{L}_t(\boldsymbol{\theta})$$
$$\Delta\boldsymbol{\theta} = -\eta\nabla_{\boldsymbol{\theta}}\mathcal{L}_t(\boldsymbol{\theta}) \tag{21}$$

where the difference between new and old network parameters are obtained using the gradients and a learning rate $\eta$. By substituting Eq. (21) into Eq. (20):

$$\nabla_{\boldsymbol{\theta}}\mathcal{L}_t(\boldsymbol{\theta})^2 = \frac{\Delta_{\boldsymbol{\theta}}\mathcal{L}_t(\boldsymbol{\theta})}{-\eta} + \epsilon(\boldsymbol{\theta}) = \frac{\mathcal{L}_t(\boldsymbol{\theta}_{\mathrm{old}}) - \mathcal{L}_t(\boldsymbol{\theta}_{\mathrm{new}})}{\eta} + \epsilon(\boldsymbol{\theta})$$

*Note the flip in sign

$$\nabla_{\boldsymbol{\theta}}\mathcal{L}_t(\boldsymbol{\theta}) = \sqrt{\frac{\Delta_{\boldsymbol{\theta}}\mathcal{L}_t(\boldsymbol{\theta})}{\eta} + \epsilon(\boldsymbol{\theta})} \approx \sqrt{\frac{\Delta_{\boldsymbol{\theta}}\mathcal{L}_t(\boldsymbol{\theta})}{\eta}} \tag{22}$$

with the small error term $\epsilon(\boldsymbol{\theta})$ dropped.

**Key Assumptions:** Our main paper highlighted the essential assumptions underlying this formulation: 1.) The loss terms $\mathcal{L}_t$ are convex and optimizable by gradient descent. 2.) Each loss term monotonically decreases i.e., the loss evaluated at previous iterations will always be strictly larger than the loss at the current iteration $\mathcal{L}_t(\boldsymbol{\theta}_{\mathrm{old}}) > \mathcal{L}_t(\boldsymbol{\theta}_{\mathrm{new}})$. This assumption ensures that the ratio $\sqrt{\frac{\Delta_{\boldsymbol{\theta}}\mathcal{L}_t(\boldsymbol{\theta})}{\eta}}$ remains positive, avoiding the computation of complex numbers. Moreover, the small learning rates commonly used in deep learning frameworks tend to be sufficiently small (e.g., $\eta = 2.5e-4$) allowing for accurate linear approximations. In practice, we can apply the ReLU function to the gradient update, i.e $\sqrt{\mathrm{ReLU}(\frac{\Delta_{\boldsymbol{\theta}}\mathcal{L}_t(\boldsymbol{\theta})}{\eta})}$ if the gradient descent step leads to an increase in the loss, violating the assumption that $\mathcal{L}_t(\boldsymbol{\theta}_{\mathrm{old}}) > \mathcal{L}_t(\boldsymbol{\theta}_{\mathrm{new}})$. This ensures that the update is scaled down or ignored in the optimization process.

**Simple Empirical Example** We further support our findings by including a simple empirical example comparing gradient descent and our proposed method. Consider a smooth, convex objective function $\mathcal{L}_{\theta} = \theta^2$. Fig. 9a illustrates our goal of obtaining a set of parameters $\theta$ such that $\mathcal{L}_{\theta}$ is minimized. For a fixed learning rate of $\eta = 0.1$, 1000 iteration steps and a starting point of $\theta = 10$, our solution given by our method (in green) is relatively close compared to the solution given by gradient descent (in blue), with a slight delay and an error of roughly 0.5. We can further improve our method's solution by reducing the learning rate to $\eta = 0.01$, which provides an even closer estimate to the solutions given by gradient descent and a reduced relative error of roughly 0.05.

## D    Supplementary Experiments and Results

### D.1    Dataset Details

**Synthetic OOD** We train our models with clean images from the original CIFAR, TinyImageNet and evaluate their OOD performance on their corrupted forms CIFAR-C, TinyImageNet-C.

1. CIFAR10/CIFAR100 (Krizhevsky & Hinton, 2009) RGB images of size (32x32) containing ten and hundred classes. The training/validation/testing sets contain 45,000/5,000/10,000 samples respectively.

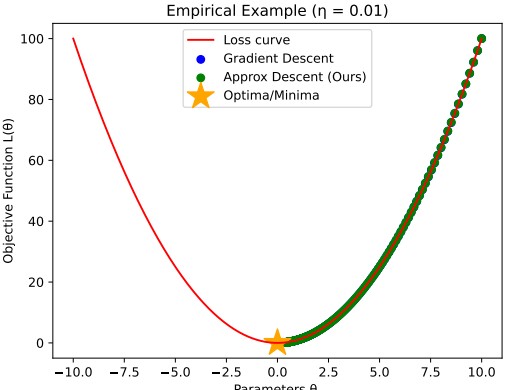
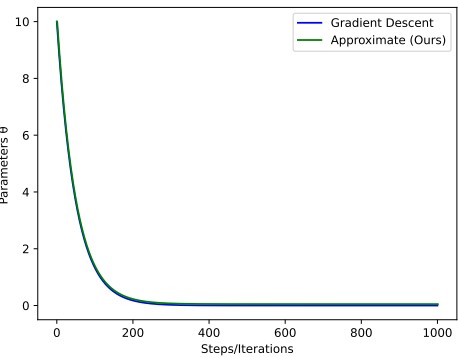

(a) By reducing the learning rate, our method (green) provides an even closer estimate to gradient descent (blue).

(b) We compare the solutions given by our method and gradient descent for $\eta = 0.01$, our method can be improved by reducing the learning rate.

2. TinyImagenet (Deng et al., 2009) A miniature version of the ImageNet dataset containing images of size (64x64) of 200 classes. There are 100,000 images for training and 10,000 images for validation/testing.

3. CIFAR10-C/CIFAR100-C/TinyImagenet-C (Hendrycks & Dietterich, 2019) A widely popular calibration benchmark, containing corrupted variants of CIFAR and TinyImageNet. Standard image corruptions (total of 19) are applied on the original test sets across five increasing levels of severities.

**Real-world OOD** For wild OOD, we learn our models using the provided ID training sets and OOD sets for validation and testing (Koh et al., 2021).

1. Camelyon17 (Bandi et al., 2019): A binary task to detect if a (32x32) cell tissue slide is benign or malignant. The images are collected across different hospitals with equipment that may vary OOD from the training set.

2. iWildCam (Beery et al., 2020): Animal species tend to vary across different backgrounds and terrains. The goal is to classify 182 animal classes collected from camera traps deployed in different areas of the wilderness.

3. FMoW (Christie et al., 2018): Satellite imagery of topographies and buildings alike tend to differ greatly across countries. The task is classify the OOD shifted terrains from one out of 62 classes.

4. CivilComments (Borkan et al., 2019): A binary text-classification task, where the model needs to identify toxic comments. The OOD shifts stem from inputs collected from differing demographics such as gender, religion, etc.

5. Amazon (Ni et al., 2019): A consumer-rating dataset where the input is a text review, with a label from a 1-to-5 star rating.

D.2 SUPPLEMENTARY EXPERIMENTS AND RESULTS

**ID Results:** As demonstrated in Table 5, our synthetic benchmark results confirm *Peacock*'s highly competitive performance on ID test sets. As discussed in the main text, *Peacock*'s calibration error is inherently bounded by the average calibration error of its constituent components. Consequently, even if some components underperform, *Peacock*'s overall calibration remains well-calibratied, irrespective of whether the data is in-distribution (ID) or out-of-distribution (OOD).

**Additional OOD Results:** Table 5, shows OOD supplementary results evaluated using AdaECE (Nguyen & O'Connor, 2015) and OE (Thulasidasan et al., 2019). Our analysis using these additional metrics aligns with the results presented in our primary findings. While the theoretical proofs in our

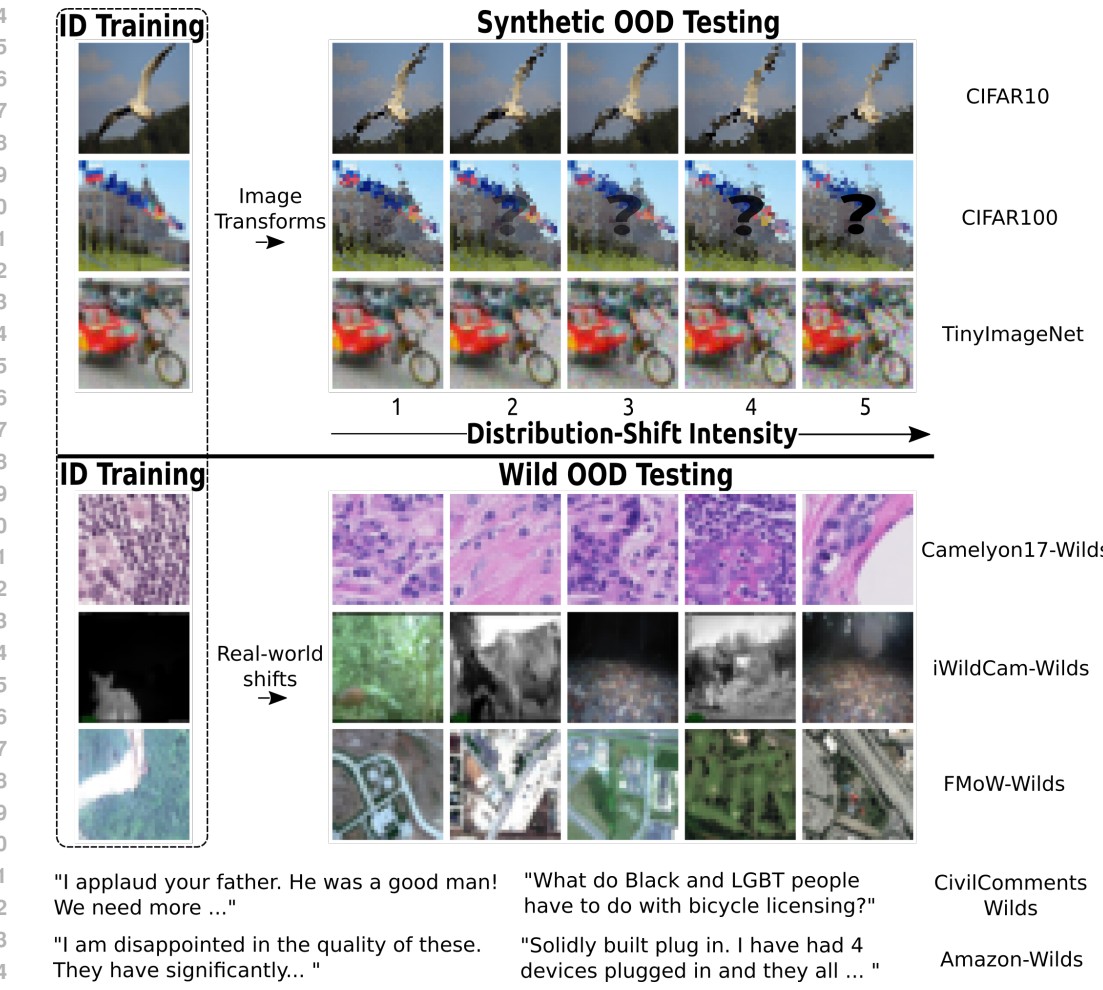

Figure 11: Covariate shifts can be simulated using common image corruptions or caused by natural differences during data collection in-the-wild.

| Dataset | Metric | MaxEnt | AdaFocal | RankMixup | CPC | Dual | ACLS | Peacock (Eq.) | Peacock (Impt.) |
|---------|--------|--------|----------|-----------|-----|------|------|---------------|-----------------|
| CIFAR10 | Acc. ↑ | $94.0_{\pm0.2}$ | $93.4_{\pm0.4}$ | $94.1_{\pm0.1}$ | $94.6_{\pm0.1}$ | $94.5_{\pm0.1}$ | $94.3_{\pm0.1}$ | $93.8_{\pm0.1}$ | $93.9_{\pm0.2}$ |
| | ECE ↓ | $1.1_{\pm0.1}$ | $0.8_{\pm0.1}$ | $3.1_{\pm0.1}$ | $2.9_{\pm0.1}$ | $1.3_{\pm0.1}$ | $3.0_{\pm0.1}$ | $\underline{0.6_{\pm0.1}}$ | $\mathbf{0.6_{\pm0.1}}$ |
| | CECE ↓ | $0.4_{\pm0.1}$ | $0.3_{\pm0.1}$ | $2.8_{\pm0.1}$ | $2.7_{\pm0.1}$ | $0.4_{\pm0.1}$ | $2.7_{\pm0.1}$ | $\underline{0.2_{\pm0.1}}$ | $\mathbf{0.2_{\pm0.1}}$ |
| | NLL ↓ | $249.8_{\pm0.4}$ | $232.7_{\pm0.1}$ | $346.9_{\pm0.2}$ | $394.4_{\pm0.2}$ | $253.7_{\pm0.1}$ | $345.3_{\pm0.3}$ | $\mathbf{224.4_{\pm0.4}}$ | $\underline{224.5_{\pm4.1}}$ |
| CIFAR100 | Acc. ↑ | $73.8_{\pm0.1}$ | $75.8_{\pm0.1}$ | $74.9_{\pm0.3}$ | $74.7_{\pm0.1}$ | $75.4_{\pm0.1}$ | $75.3_{\pm0.1}$ | $74.5_{\pm0.4}$ | $73.5_{\pm0.5}$ |
| | ECE ↓ | $5.4_{\pm0.5}$ | $6.8_{\pm0.1}$ | $4.9_{\pm0.1}$ | $8.8_{\pm0.3}$ | $9.1_{\pm0.1}$ | $4.5_{\pm0.3}$ | $\underline{4.1_{\pm0.3}}$ | $\mathbf{3.9_{\pm0.2}}$ |
| | CECE ↓ | $0.2_{\pm0.1}$ | $0.1_{\pm0.1}$ | $2.9_{\pm0.1}$ | $2.3_{\pm0.2}$ | $0.1_{\pm0.1}$ | $1.7_{\pm0.1}$ | $\underline{0.1_{\pm0.1}}$ | $\mathbf{0.1_{\pm0.1}}$ |
| | NLL ↓ | $312.7_{\pm0.6}$ | $306.5_{\pm2.3}$ | $348.6_{\pm0.7}$ | $432.2_{\pm0.8}$ | $319.8_{\pm0.4}$ | $346.6_{\pm1.0}$ | $\underline{298.3_{\pm1.2}}$ | $\mathbf{283.9_{\pm0.1}}$ |
| TinyImageNet | Acc. ↑ | $63.1_{\pm0.3}$ | $60.8_{\pm0.1}$ | $61.6_{\pm0.3}$ | $65.0_{\pm0.3}$ | $63.2_{\pm0.3}$ | $64.9_{\pm0.1}$ | $61.2_{\pm0.1}$ | $62.3_{\pm0.4}$ |
| | ECE ↓ | $18.2_{\pm0.3}$ | $6.1_{\pm0.5}$ | $\underline{5.5_{\pm0.3}}$ | $10.3_{\pm0.4}$ | $6.8_{\pm0.1}$ | $5.0_{\pm0.3}$ | $6.2_{\pm0.3}$ | $\mathbf{3.9_{\pm0.3}}$ |
| | CECE ↓ | $0.1_{\pm0.1}$ | $0.1_{\pm0.1}$ | $0.1_{\pm0.1}$ | $0.1_{\pm0.1}$ | $0.1_{\pm0.1}$ | $0.1_{\pm0.1}$ | $\underline{0.1_{\pm0.1}}$ | $\mathbf{0.1_{\pm0.1}}$ |
| | NLL ↓ | $322.0_{\pm0.5}$ | $320.5_{\pm0.3}$ | $343.2_{\pm1.0}$ | $358.5_{\pm1.6}$ | $339.2_{\pm1.0}$ | $342.3_{\pm0.4}$ | $\underline{333.9_{\pm2}}$ | $\mathbf{324.2_{\pm2.4}}$ |

Table 5: We report the ID test scores (%) for reruns computed across 3 seeds for *Peacock* and its components.

main paper and Appendix C are explicitly stated only for the ECE, we anticipate that our arguments remain valid for other calibration metrics, which are often derivatives or closely related to ECE. We intend to explore this aspect further in future research.

**Additional Multi-Objective Optimization Results:** Additional results for our proposed weighted-importance formulation are provided in Table 8 and Table 9. Our results highlight the versatility, effectiveness and speed acorss a wide variety of different architectures and methods.

| Dataset | Metric | MaxEnt | AdaFocal | RankMixup | CPC | Dual | ACLS | Peacock (Eq.) | Peacock (Impt.) |
|---|---|---|---|---|---|---|---|---|---|
| CIFAR10-C | AdaECE ↓ | $6.9_{\pm0.1}$ | $6.2_{\pm0.4}$ | $11.5_{\pm0.2}$ | $10.7_{\pm0.4}$ | $7.2_{\pm0.2}$ | $11.5_{\pm0.1}$ | $6.3_{\pm0.1}$ | $\mathbf{6.2}_{\pm0.3}$ |
| | OE ↓ | $3.9_{\pm0.3}$ | $\mathbf{3.0}_{\pm0.3}$ | $9.6_{\pm0.2}$ | $9.1_{\pm0.4}$ | $3.8_{\pm0.1}$ | $9.5_{\pm0.1}$ | $3.3_{\pm0.1}$ | $3.5_{\pm0.2}$ |
| CIFAR100-C | AdaECE ↓ | $11.0_{\pm0.1}$ | $13.7_{\pm0.3}$ | $8.4_{\pm0.2}$ | $13.7_{\pm0.2}$ | $15.5_{\pm0.1}$ | $10.2_{\pm0.3}$ | $9.7_{\pm0.3}$ | $\mathbf{9.6}_{\pm0.3}$ |
| | OE ↓ | $\mathbf{0.5}_{\pm0.1}$ | $0.7_{\pm0.1}$ | $2.5_{\pm0.2}$ | $1.8_{\pm0.1}$ | $0.7_{\pm0.1}$ | $2.01_{\pm0.1}$ | $1.3_{\pm0.1}$ | $1.6_{\pm0.1}$ |
| TinyImageNet-C | AdaECE ↓ | $12.6_{\pm0.3}$ | $13.9_{\pm0.3}$ | $20.2_{\pm0.2}$ | $16.3_{\pm0.2}$ | $18.8_{\pm0.2}$ | $20.6_{\pm0.4}$ | $\mathbf{10.4}_{\pm0.2}$ | $10.7_{\pm0.2}$ |
| | OE ↓ | $4.4_{\pm0.3}$ | $4.5_{\pm0.3}$ | $10.4_{\pm0.2}$ | $8.5_{\pm0.2}$ | $9.4_{\pm0.2}$ | $11.1_{\pm0.4}$ | $\mathbf{2.3}_{\pm0.2}$ | $2.3_{\pm0.2}$ |
| Camelyon17 | AdaECE ↓ | $12.3_{\pm0.4}$ | $20.4_{\pm0.1}$ | $22.4_{\pm4.6}$ | $20.1_{\pm1.6}$ | $15.4_{\pm2.5}$ | $19.6_{\pm0.2}$ | $11.7_{\pm0.7}$ | $\mathbf{9.8}_{\pm1.8}$ |
| | OE ↓ | $10.9_{\pm0.8}$ | $19.6_{\pm0.1}$ | $22.0_{\pm4.6}$ | $19.8_{\pm1.6}$ | $14.3_{\pm2.4}$ | $18.9_{\pm0.1}$ | $10.6_{\pm0.4}$ | $\mathbf{9.0}_{\pm1.6}$ |
| iWildCam | AdaECE ↓ | $21.0_{\pm3.2}$ | $23.0_{\pm0.5}$ | $25.5_{\pm0.7}$ | $20.3_{\pm1.1}$ | $13.0_{\pm2.5}$ | $20.6_{\pm1.8}$ | $\mathbf{9.7}_{\pm0.3}$ | $12.6_{\pm1.4}$ |
| | OE ↓ | $14.3_{\pm3.6}$ | $16.8_{\pm0.2}$ | $20.4_{\pm0.8}$ | $15.5_{\pm0.2}$ | $8.21_{\pm1.4}$ | $15.4_{\pm1.2}$ | $\mathbf{5.2}_{\pm0.2}$ | $7.9_{\pm1.0}$ |
| FmoW | AdaECE ↓ | $20.0_{\pm9.9}$ | $20.9_{\pm8.6}$ | $41.7_{\pm0.1}$ | $22.4_{\pm0.9}$ | $\mathbf{9.73}_{\pm0.1}$ | $21.7_{\pm0.2}$ | $10.5_{\pm0.2}$ | $10.6_{\pm0.1}$ |
| | OE ↓ | $13.7_{\pm7.7}$ | $14.2_{\pm7.0}$ | $33.7_{\pm0.2}$ | $16.7_{\pm0.8}$ | $\mathbf{4.78}_{\pm0.1}$ | $14.6_{\pm0.2}$ | $5.5_{\pm0.3}$ | $5.4_{\pm0.3}$ |

Table 6: We report additional OOD test scores (%) for reruns evaluated on both synthetic and wild benchmarks for *Peacock* and its components.

| Dataset | ECE ↓ | CE | MaxEnt | AdaFocal | RankMixup | CPC | Dual | ACLS | Peacock (Eq.) | Peacock (Impt.) |
|---|---|---|---|---|---|---|---|---|---|---|
| Amazon | Pre | $7.0_{\pm0.5}$ | $5.0_{\pm0.6}$ | $6.7_{\pm1.0}$ | $43.1_{\pm0.1}$ | $7.4_{\pm0.5}$ | $5.8_{\pm2.3}$ | $42.0_{\pm0.1}$ | $6.5_{\pm3.2}$ | $\mathbf{5.0}_{\pm1.1}$ |
| | Post | $11.6_{\pm0.5}$ | $7.6_{\pm0.4}$ | $10.0_{\pm0.3}$ | $41.0_{\pm0.2}$ | $3.4_{\pm1.2}$ | $5.0_{\pm1.5}$ | $26.3_{\pm0.1}$ | $5.6_{\pm1.7}$ | $\mathbf{4.8}_{\pm0.2}$ |
| | Avg. | $9.3_{\pm0.5}$ | $6.3_{\pm0.2}$ | $8.4_{\pm0.2}$ | $42.0_{\pm0.1}$ | $5.4_{\pm0.3}$ | $5.4_{\pm0.2}$ | $34.2_{\pm0.3}$ | $6.1_{\pm0.3}$ | $\mathbf{4.9}_{\pm0.3}$ |
| | Temp. | 1.50 | 1.25 | 1.25 | 1.25 | 1.25 | 1.25 | 1.25 | 1.25 | 1.25 |
| CivilComments | Pre | $10.4_{\pm0.4}$ | $4.8_{\pm0.2}$ | $7.8_{\pm1.7}$ | $11.4_{\pm0.5}$ | $2.4_{\pm0.4}$ | $4.2_{\pm0.1}$ | $11.1_{\pm0.1}$ | $\mathbf{2.1}_{\pm0.8}$ | $4.2_{\pm1.0}$ |
| | Post | $6.3_{\pm0.9}$ | $8.2_{\pm0.7}$ | $11.6_{\pm0.2}$ | $11.0_{\pm0.1}$ | $2.2_{\pm0.3}$ | $7.5_{\pm0.4}$ | $6.7_{\pm0.1}$ | $\mathbf{5.7}_{\pm0.7}$ | $7.5_{\pm0.5}$ |
| | Avg. | $8.4_{\pm0.6}$ | $6.5_{\pm0.2}$ | $9.7_{\pm1.5}$ | $11.2_{\pm0.5}$ | $2.3_{\pm0.4}$ | $5.9_{\pm0.1}$ | $8.9_{\pm0.2}$ | $\mathbf{3.9}_{\pm0.9}$ | $5.8_{\pm0.9}$ |
| | Temp. | 2.00 | 1.25 | 1.25 | 1.25 | 1.25 | 1.25 | 1.25 | 1.25 | 1.25 |

Table 7: Vanilla temperature scaling results with temperatures obtained post-grid search for Wilds-Text datasets.

| Algorithm | Acc (%) | ECE (%) | Speed (Sec) | w1 | w2 | w3 | $\sum_t^A w_t = 1$ |
|---|---|---|---|---|---|---|---|
| Equal-Importance | $76.8_{\pm0.2}$ | $6.3_{\pm0.1}$ | $48.3_{\pm0.2}$ | 0.33 | 0.33 | 0.33 | Yes |
| MTAN (Liu et al., 2019a) | $76.5_{\pm0.1}$ | $6.5_{\pm0.1}$ | $48.5_{\pm0.2}$ | 0.33 | 0.33 | 0.33 | Yes |
| CoVV (Groenendijk et al., 2021) | $77.3_{\pm0.1}$ | $6.5_{\pm0.1}$ | $48.9_{\pm0.1}$ | 0.01 | 0.52 | 0.47 | Yes |
| GradNorm (Chen et al., 2018) | $75.7_{\pm0.6}$ | $6.8_{\pm0.4}$ | $79.1_{\pm0.1}$ | 2.99 | 0.00 | 0.00 | No |
| MT-MOO (Sener & Koltun, 2018) | $76.4_{\pm0.4}$ | $6.5_{\pm0.1}$ | $66.3_{\pm0.3}$ | 0.36 | 0.47 | 0.16 | Yes |
| Weighted-Importance (Ours) | $77.3_{\pm0.4}$ | $\mathbf{6.2}_{\pm0.3}$ | $48.5_{\pm0.2}$ | 0.00 | 0.53 | 0.47 | Yes |

Table 8: Comparisons of different multi-objective optimization methods for *Peacock* evaluated on CIFAR10/CIFAR10-C using ResNet-18.

| Algorithm (CIFAR10-C) | Acc (%) | ECE (%) | Speed (Sec) | w1 | w2 | w3 | $\sum_t w_t = 1$ |
|---|---|---|---|---|---|---|---|
| Equal-Importance | $82.7_{\pm0.1}$ | $9.8_{\pm0.3}$ | $838_{\pm3}$ | 0.33 | 0.33 | 0.33 | Yes |
| CoVV (Groenendijk et al., 2021) | $83.1_{\pm0.2}$ | $8.5_{\pm0.3}$ | $827_{\pm5}$ | 0.02 | 0.22 | 0.76 | Yes |
| GradNorm (Chen et al., 2018) | $80.2_{\pm0.4}$ | $9.7_{\pm0.4}$ | $1347_{\pm3}$ | 3.00 | 0.00 | 0.00 | No |
| MT-MOO (Sener & Koltun, 2018) | $80.7_{\pm0.1}$ | $9.8_{\pm0.1}$ | $915_{\pm3}$ | 0.43 | 0.54 | 0.03 | Yes |
| Weighted-Importance (Ours) | $81.0_{\pm0.4}$ | $\mathbf{6.1}_{\pm0.3}$ | $840_{\pm5}$ | 0.00 | 0.53 | 0.47 | Yes |

Table 9: Comparisons of different multi-objective methods for *Peacock* using SWINV2.

## D.3 ABLATION STUDIES

To gain a better understanding of each component in *Peacock*, we provide an ablation study that removes each component from the full combination of *Peacock*. In Table 10, we show the respective ECE, OE and KSE scores of each combination evaluated on CIFAR/CIFAR-C. While each component generally helps improve calibration performance, we identify RankMixup and MaxEnt loss as two of the most critical building blocks of *Peacock*. Since the removal of either RankMixup or MaxEnt loss would cause a noticeable drop in calibration performance. Although ACLS independently delivers competitive performance, we find it to be the least impactful, since its removal leads to better calibration in *Peacock*. Therefore we propose the final version of equal and importance weighted forms *Peacock* to be without ACLS. Note that the experiments performed in this ablation study does not include temperature scaling. For e.g., removing RankMixup, would cause the highest ECE on CIFAR10/CIFAR10-C with 1.9% and 10.1% respectively. The lack of MaxEnt loss constraints delivers the worst result on CIFAR100/CIFAR100-C with 8.4% and 15.3%.

| Algorithm (ID Performance) | (a) CIFAR10 | | | (b) CIFAR100 | | |
|---|---|---|---|---|---|---|
| | ECE | NLL | KSE | ECE | NLL | KSE |
| Peacock w/o MaxE | $1.2_{\pm 0.1}$ | $271.8_{\pm 2.9}$ | $1.1_{\pm 0.1}$ | $8.4_{\pm 0.1}$ | $316.8_{\pm 1.7}$ | $8.4_{\pm 0.1}$ |
| Peacock w/o AdaFocal | $1.7_{\pm 0.1}$ | $289.5_{\pm 4.2}$ | $1.8_{\pm 0.1}$ | $6.1_{\pm 0.7}$ | $314.8_{\pm 1.0}$ | $6.1_{\pm 0.7}$ |
| Peacock w/o RankMixup | $1.9_{\pm 0.2}$ | $294.8_{\pm 3.2}$ | $2.0_{\pm 0.2}$ | $6.1_{\pm 0.3}$ | $316.6_{\pm 0.9}$ | $6.2_{\pm 0.3}$ |
| Peacock w/o CPC | $1.6_{\pm 0.2}$ | $284.6_{\pm 6.3}$ | $1.9_{\pm 0.2}$ | $\mathbf{6.0}_{\pm 0.7}$ | $317.8_{\pm 1.9}$ | $\mathbf{6.0}_{\pm 0.7}$ |
| Peacock w/o Dual | $1.5_{\pm 0.1}$ | $278.2_{\pm 2.9}$ | $1.7_{\pm 0.1}$ | $6.5_{\pm 0.2}$ | $\mathbf{308.6}_{\pm 0.9}$ | $6.5_{\pm 0.2}$ |
| Peacock w/o ACLS | $\mathbf{0.6}_{\pm 0.1}$ | $\mathbf{240.9}_{\pm 0.4}$ | $\mathbf{0.9}_{\pm 0.1}$ | $6.5_{\pm 0.2}$ | $306.3_{\pm 0.9}$ | $6.8_{\pm 0.2}$ |
| Algorithm (OOD Performance) | (a) CIFAR10-C | | | (b) CIFAR100-C | | |
| | ECE | NLL | KSE | ECE | NLL | KSE |
| Peacock w/o MaxE | $7.6_{\pm 0.4}$ | $270.4_{\pm 2.9}$ | $7.2_{\pm 0.4}$ | $15.3_{\pm 0.1}$ | $360.0_{\pm 0.5}$ | $14.9_{\pm 0.1}$ |
| Peacock w/o AdaFocal | $8.6_{\pm 0.4}$ | $286.0_{\pm 1.3}$ | $8.3_{\pm 0.4}$ | $12.4_{\pm 0.6}$ | $355.9_{\pm 1.3}$ | $12.2_{\pm 0.6}$ |
| Peacock w/o RankMixup | $10.1_{\pm 0.4}$ | $296.8_{\pm 2.8}$ | $9.9_{\pm 0.5}$ | $12.7_{\pm 0.4}$ | $358.4_{\pm 0.4}$ | $12.4_{\pm 0.4}$ |
| Peacock w/o CPC | $8.7_{\pm 0.4}$ | $285.0_{\pm 1.8}$ | $8.3_{\pm 0.3}$ | $12.4_{\pm 0.7}$ | $359.9_{\pm 1.7}$ | $12.3_{\pm 0.7}$ |
| Peacock w/o Dual | $8.0_{\pm 0.1}$ | $278.7_{\pm 0.8}$ | $7.7_{\pm 0.1}$ | $12.5_{\pm 0.2}$ | $\mathbf{353.2}_{\pm 1.1}$ | $12.3_{\pm 0.2}$ |
| Peacock w/o ACLS | $\mathbf{6.5}_{\pm 0.2}$ | $\mathbf{245.6}_{\pm 0.4}$ | $\mathbf{6.3}_{\pm 0.1}$ | $\mathbf{11.6}_{\pm 0.3}$ | $358.3_{\pm 0.5}$ | $\mathbf{11.7}_{\pm 0.5}$ |

Table 10: Component analysis of *Peacock* reveals the best performance when all algorithms except ACLS are combined.

# E    LIMITATIONS

**Component Permutations**    In the case of *Peacock*, we featured a total of seven baselines which gives a total of $2^7 - 1$ permutations. While the primary focus of our paper is looking at whether different calibration algorithms can be successfully combined, we constrained *Peacock* to the seven featured algorithms so as to keep experiments manageable. We note that there are many potential algorithms in the calibration family that could become promising candidates (see Fig. 2a).

**Modularity and Future Components**    To the best of our ability, we built *Peacock* based on the most relevant SOTA calibration components. For each algorithm, we closely referenced the source code provided by the respective authors. As we believe that *Peacock* will perform as well/better than the average of its components, we specifically built *Peacock* in a modular fashion allowing the easy integration of future methods.

# F REPRODUCIBILITY CHECKLIST

If needed, we provide the reproducibility checklist of this paper.

This paper:

- Includes a conceptual outline and/or pseudocode description of AI methods introduced (yes)
- Clearly delineates statements that are opinions, hypothesis, and speculation from objective facts and results (yes)
- Provides well marked pedagogical references for less-familiare readers to gain background necessary to replicate the paper (yes)

Does this paper make theoretical contributions? (yes)

If yes, please complete the list below.

- All assumptions and restrictions are stated clearly and formally. (yes)
- All novel claims are stated formally (e.g., in theorem statements). (yes)
- Proofs of all novel claims are included. (yes)
- Proof sketches or intuitions are given for complex and/or novel results. (yes)
- Appropriate citations to theoretical tools used are given. (yes)
- All theoretical claims are demonstrated empirically to hold. (yes)
- All experimental code used to eliminate or disprove claims is included. (yes)

Does this paper rely on one or more datasets? (yes)

If yes, please complete the list below.

- A motivation is given for why the experiments are conducted on the selected datasets (yes)
- All novel datasets introduced in this paper are included in a data appendix. (NA)
- All novel datasets introduced in this paper will be made publicly available upon publication of the paper with a license that allows free usage for research purposes. (NA)
- All datasets drawn from the existing literature (potentially including authors' own previously published work) are accompanied by appropriate citations. (yes)
- All datasets drawn from the existing literature (potentially including authors' own previously published work) are publicly available. (yes)
- All datasets that are not publicly available are described in detail, with explanation why publicly available alternatives are not scientifically satisficing. (NA)

Does this paper include computational experiments? (yes)

If yes, please complete the list below.

- Any code required for pre-processing data is included in the appendix. (yes).
- All source code required for conducting and analyzing the experiments is included in a code appendix. (yes)
- All source code required for conducting and analyzing the experiments will be made publicly available upon publication of the paper with a license that allows free usage for research purposes. (yes)
- All source code implementing new methods have comments detailing the implementation, with references to the paper where each step comes from (yes)

- If an algorithm depends on randomness, then the method used for setting seeds is described in a way sufficient to allow replication of results. (yes)

- This paper specifies the computing infrastructure used for running experiments (hardware and software), including GPU/CPU models; amount of memory; operating system; names and versions of relevant software libraries and frameworks. (yes)

- This paper formally describes evaluation metrics used and explains the motivation for choosing these metrics. (yes)

- This paper states the number of algorithm runs used to compute each reported result. (yes)

- Analysis of experiments goes beyond single-dimensional summaries of performance (e.g., average; median) to include measures of variation, confidence, or other distributional information. (yes)

- The significance of any improvement or decrease in performance is judged using appropriate statistical tests (e.g., Wilcoxon signed-rank). (yes)

- This paper lists all final (hyper-)parameters used for each model/algorithm in the paper's experiments. (yes)

- This paper states the number and range of values tried per (hyper-) parameter during development of the paper, along with the criterion used for selecting the final parameter setting. (yes)

