| $1.2_{\pm0.1}$ | $271.8_{\pm2.9}$ | $1.1_{\pm0.1}$ | $8.4_{\pm0.1}$ | $316.8_{\pm1.7}$ | $8.4_{\pm0.1}$ |
| Peacock w/o AdaFocal | $1.7_{\pm0.1}$ | $289.5_{\pm4.2}$ | $1.8_{\pm0.1}$ | $6.1_{\pm0.7}$ | $314.8_{\pm1.0}$ | $6.1_{\pm0.7}$ |
| Peacock w/o RankMixup | $1.9_{\pm0.2}$ | $294.8_{\pm3.2}$ | $2.0_{\pm0.2}$ | $6.1_{\pm0.3}$ | $316.6_{\pm0.9}$ | $6.2_{\pm0.3}$ |
| Peacock w/o CPC | $1.6_{\pm0.2}$ | $284.6_{\pm6.3}$ | $1.9_{\pm0.2}$ | $\mathbf{6.0}_{\pm0.7}$ | $317.8_{\pm1.9}$ | $\mathbf{6.0}_{\pm0.7}$ |
| Peacock w/o Dual | $1.5_{\pm0.1}$ | $278.2_{\pm2.9}$ | $1.7_{\pm0.1}$ | $6.5_{\pm0.2}$ | $\mathbf{308.6}_{\pm0.9}$ | $6.5_{\pm0.2}$ |
| Peacock w/o ACLS | $\mathbf{0.6}_{\pm0.1}$ | $\mathbf{240.9}_{\pm0.4}$ | $\mathbf{0.9}_{\pm0.1}$ | $6.5_{\pm0.2}$ | $306.3_{\pm0.9}$ | $6.8_{\pm0.2}$ |
| | (a) CIFAR10-C | | | (b) CIFAR100-C | | |
| Algorithm (OOD Performance) | ECE | NLL | KSE | ECE | NLL | KSE |
| Peacock w/o MaxE | $7.6_{\pm0.4}$ | $270.4_{\pm2.9}$ | $7.2_{\pm0.4}$ | $15.3_{\pm0.1}$ | $360.0_{\pm0.5}$ | $14.9_{\pm0.1}$ |
| Peacock w/o AdaFocal | $8.6_{\pm0.4}$ | $286.0_{\pm1.3}$ | $8.3_{\pm0.4}$ | $12.4_{\pm0.6}$ | $355.9_{\pm1.3}$ | $12.2_{\pm0.6}$ |
| Peacock w/o RankMixup | $10.1_{\pm0.4}$ | $296.8_{\pm2.8}$ | $9.9_{\pm0.5}$ | $12.7_{\pm0.4}$ | $358.4_{\pm0.4}$ | $12.4_{\pm0.4}$ |
| Peacock w/o CPC | $8.7_{\pm0.4}$ | $285.0_{\pm1.8}$ | $8.3_{\pm0.3}$ | $12.4_{\pm0.7}$ | $359.9_{\pm1.7}$ | $12.3_{\pm0.7}$ |
| Peacock w/o Dual | $8.0_{\pm0.1}$ | $278.7_{\pm0.8}$ | $7.7_{\pm0.1}$ | $12.5_{\pm0.2}$ | $\mathbf{353.2}_{\pm1.1}$ | $12.3_{\pm0.2}$ |
| Peacock w/o ACLS | $\mathbf{6.5}_{\pm0.2}$ | $\mathbf{245.6}_{\pm0.4}$ | $\mathbf{6.3}_{\pm0.1}$ | $\mathbf{11.6}_{\pm0.3}$ | $358.3_{\pm0.5}$ | $\mathbf{11.7}_{\pm0.5}$ |