# OpenReview forum: "Peacock: Multi-Objective Optimization for Deep Neural Network Calibration"
_ICLR.cc/2025/Conference — Submitted to ICLR 2025_

### Official Review · Reviewer_jFTV · 2024-10-30

**Soundness:** 2
**Presentation:** 3
**Contribution:** 3
**Rating:** 5
**Confidence:** 4

**Summary:**

The paper proposes a multi-objective calibration strategy that relies on Pareto optimal solutions that combines several losses that have been shown to provide good calibrated networks. The approach is evaluated on several small size problems (image and text).

**Strengths:**

- The principle of combining multiple losses designed for good calibration makes sense.

- Comprehensive evaluation and ablation study on multiple architectures and datasets.

- The results are constantly good on the evaluated problems (but only small size problems).

I rate the work as “marginally below the acceptance threshold”: I may change my score after reading the rebuttal and other reviewers' comments. .

**Weaknesses:**

- The proposed approach is basically a multi-objective optimization using the latest calibration losses from the literature: the algorithm novelty is limited.

- I had a problem with the mathematical notations which are often not clearly defined. This makes the paper difficult to check. See the questions section.

- Evaluation is only for small size problems (according to current standards).

**Questions:**

- I had problems with the mathematical notations. For example (but not only):
   - l.139 and 141: is $P_i(y_k | x)=P(\{y=y_k\}|x_i)$ the likelihood or score of hypothesis $k$ when $x_i$ is the input ? What is $h_i$?
  - Eq. 1: ECE is a scalar measure that characterizes globally the predictor (Naeini et al., 2015): How can it depend on a sample $x$?
  - l. 281: what is $\bar{y}$?
  - Eq.11:  I don’t understand this equation (expectation of a non random scalar).
  - Please make your mathematical notations sound.

- How does your approach scale with larger models and datasets (ImageNet) ?

- How does the approach compare with post-hoc calibration, which is a much lighter calibration strategy, for instance [1] cited in the paper or [2] not cited?

[1] Jize Zhang, Bhavya Kailkhura, and T Yong-Jin Han. Mix-n-match: Ensemble and compositional methods
for uncertainty calibration in deep learning. In ICML, 2020.

[2] Kanil Patel, William H Beluch, Bin Yang, Michael Pfeiffer, and Dan Zhang. Multi-class uncertainty
calibration via mutual information maximization-based binning. In ICLR, 2021

---

> ### Author Response · Authors · 2024-11-23
> **Rebuttal**
>
> ### Reviewer jFTV
> > Q: Algorithm novelty
>
> We thank the reviewer for this concern. While the key idea of multi-objective optimization of calibration strategies may seem intuitive, our work introduces several key innovations that significantly contribute to the field of model calibration:
>
> 1. Our paper highlights the underlying issues in deep neural network calibration such as overconfidence and OOD shifts and pays tribute to all recent contributions.
> 2. We are the first work that introduces a multi-objective framework for neural network calibration, with both and equal and importance weighted formulation.
> 3. We propose an efficient algorithm using direction weighted self-attention to compute Pareto optimal importance weights, addressing compute overheads - which is crucial for practical deployment.
>
> > Q: Mathematical notations
>
> For our work, the variable $k$ denotes the classes. The hypothesis/logits are represented with $h(x_i)$ for a single input $x_i$.
> We have rewritten Equation 1 for clarity. $\bar{P}(h_b^{\boldsymbol{\theta}})$ denotes the averaged predicted confidence and $\bar{y_b}$ is the averaged ground truth for all $b$ samples $x$ in each bin. Equation 11 calculates the expected squared calibration error across all input samples for each hypothesis.
>
> > Q: How does your approach scale with larger models and datasets (ImageNet) ?
>
> We have no concerns over scalability as Peacock is designed to be competetive across different archictecture and datasets. For example, Table 9 shows that our method scales well on SWINV2, with highly competitive run times.
>
> > Q: How does the approach compare with post-hoc calibration
>
> In Table 2, we have included experiments with post-hoc calibration, using our choice of AdaTS. We acknowledge that there are many variants of post-hoc strategies and fundamentally believe that with some adaptation, Peacock can complement them.
>
> > Q: Citations and other works
>
> We acknowledge the contributions of the cited works and will include these references in the paper.
> We believe that our proposed approach offers a useful guide to the calibration problem. We are committed to improving the clarity of our presentation and addressing any remaining concerns.

---

> > ### Comment · Reviewer_jFTV · 2024-11-25
> > **Comment on authors' rebuttal**
> >
> > Thank you for providing answers to my questions.
> >
> > Regarding novelty, I agree that the approach is reasonably studied and shows good, consistent performance with practical implementation, which I consider a notable achievement.
> >
> > However, I cannot say that I am fully satisfied by the answers to my other questions:
> >
> > - Mathematical notations: the definition of ECE is still not good for me because the symbols used are not fully defined, for instance, $h_b^\theta(x)$. It may appear as a detail because people interested in deep network calibration know what ECE is, but it does seem difficult to me to find sound notations shared by the community. Eq.11 is still not understandable to me: ECE is not a random variable - it’s a scalar estimated from data - and I don’t see what the meaning of the expectation is in this case.
> >
> > - Larger problems: you have evaluated your approach on a transformer-like model but not on large datasets such as Imagenet, which is now customary in calibration papers. Using a combination of losses as your approach proposes may not be scalable.
> >
> > - Post-hoc calibration: my question was simply to have comparative information on two different calibration strategies, a post-hoc approach being much lighter but possibly less efficient. I do not have the answer in your response.

---

> ### Author Response · Authors · 2024-11-26
> **Rebuttal**
>
> ### Reviewer jFTV
>
> We thank the reviewer for their timely response and for acknowledging the novelty of our approach. In this rebuttal we further clarify remaining question of our work.
>
> > Q: Mathematical notations
>
> As the reviewer correctly points out, there are differing definitions of ECE in the literature. To ensure clarity, we adopt the definition presented in Equation 3 of Zhang et al. [1] in the revised manuscript. For Equation 11, the random variable is $x$ and not ECE. We estimate the empirical ECE across each hypothesis for every observed sample $x$ randomly drawn from a prior distribution $p(x)$. We thank the reviewer as this may not seem entirely obvious and further clarify this in our paper.
>
> > Q: Scalability
>
> The core idea of Peacock is independent of dataset or model size, making it scalable to large-scale models and datasets. While we currently do not have experiments conducted on ImageNet, our experiments on smaller scale problems indicate no performance degradation with increasing model or dataset size. We understand the reviewer's concern regarding scalabilty issues, since Peacock encompasses the strength of multiple algorithms. However a key contribution of our work is in addressing computational overhead issues during the ensemble of multiple losses. For example, Figure 9 shows vanilla RankMixup requiring twice the training time for CIFAR10, which we have resolved in our proposed formulation in Appendix B.2. This resolution of the compute overhead gives Peacock the same training speed as with regular cross-entropy loss. Furthermore, our importance weighted formulation in Table 3 also shows highly competetive runtimes as compared to other multi-objective optimization methods. To the best of our ability, our experiments do not indicate scalability issues across larger achitectures/datasets.
>
>
> >Q: Post-hoc calibration:
>
> While we currently do not have direct comparisons against [1][2], Section 5.3 of our main paper is dedicated towards comparisons on post-hoc processing using our choice of Adaptive Temperature scaling [3] (AdaTS). Although AdaTS cannot be considered a direct replacement for [1], we believe that this is a reasonable choice given that it is a newer method and shares the same principle of using temperature scaling. Table 2 shows the results of Peacock and it's individual components before and after AdaTS, with the first column of Table 2 indicating the results of only cross-entropy loss + AdaTS (which is the simplest formulation).
>
> * Key finding 1: Although AdaTS generally improves ECE, we can see that applying post-hoc methods to well-calibrated models can further degrade performance. This is because of the temperatures of AdaTS were obtained on a ID validation set, explaining the higher calibration errors after temperature scaling [4].
>
> * Key finding 2: In contrast, our experiments indicate Peacock having the best performance, both before and after temperature scaling, with a slight performance edge given to the importance-weighted formulation.
>
> * In general, there are both pros and cons to various calibration approaches. For example, post-hoc methods are lighter in nature and do not require retraining, however finding a suitable set of temperatures can be incredibly challenging for OOD scenarios. In contrast, calibration loss functions require retraining but provide better entropy and robustness OOD. Our work demonstrates that the loss-ensembles of multiple recent methods in Peacock can offer added benefits in performance, with an optional choice for further using post-hoc processing.
>
> Once again, we thank the reviewer for their response and welcome any further questions in the discussion phase.
>
> * [1] Jize Zhang, Bhavya Kailkhura, and T Yong-Jin Han. Mix-n-match: Ensemble and compositional methods for uncertainty calibration in deep learning. In ICML, 2020.
> * [2] Kanil Patel, William H Beluch, Bin Yang, Michael Pfeiffer, and Dan Zhang. Multi-class uncertainty calibration via mutual information maximization-based binning. In ICLR, 2021
> * [3] Tom Joy, Francesco Pinto, Ser-Nam Lim, Philip H.S. Torr, and Puneet K. Dokania. Sample-dependent adaptive temperature scaling for improved calibration. In AAAI, 2023
> * [4] Yaniv Ovadia, Emily Fertig, Jie Ren, Zachary Nado, D. Sculley, Sebastian Nowozin,
> Joshua Dillon, Balaji Lakshminarayanan, and Jasper Snoek. Can you trust your model's uncertainty? Evaluating predictive uncertainty under dataset shift. In NeuRIPS, 2019

---

> > ### Author Response · Authors · 2024-11-28
> > **Kindly let us know if you have anymore questions**
> >
> > Dear Reviewer jFTV,
> >
> > We thank you again for your valuable time in reviewing. As we believe we have engaged in the review adequately, we are willing to further answer any more questions and concerns in the discussion phase.
> >
> > We thank you again in reviewing and helping us to improve our work.
> >
> > Please kindly let us know your concerns and if further clarifications are needed.
> >
> > Sincerely,
> > Authors

---

> > ### Comment · Reviewer_jFTV · 2024-11-29
> > **Final comments**
> >
> > Thank you for your detailed answer. Here are my final remarks.
> >
> > - Mathematical notations. I am sorry to say that I still do not understand how ECE, as defined in the main article and the appendix and is a distribution-dependent scalar measure, depends on a sample $x$, and therefore requires an expectation in Eq. 11. I may miss something.
> >
> > - Scalability. Your arguments can be received, but the simplest would have been demonstrating it.
> >
> > - Post-hoc calibration. I understand that implementing another method is costly, but your statements could have been justified better by taking actual figures from the literature, for instance.
> >
> > I keep my rating as not completely satisfied by the answers.

---

> > > ### Author Response · Authors · 2024-11-30
> > > **Rebuttal**
> > >
> > > We thank the reviewer for the added comments. We address certain misunderstandings and final concerns below:
> > >
> > > > Notations
> > >
> > > We believe our notations are clear and that our previous rebuttals have addressed this. The ECE is inherently tied to the probability distribution of the model's predictions. Which means that the ECE value can vary depending on the characteristics of the data distribution. While ECE is a global measure, it's calculated based on the aggregate performance of each hypothesis across many samples, which is why the expecation is taken across the random variables $x$. As the other reviewers have no issues with the notations, could you please provide more specific feedback about which part you are unsure?
> > >
> > > > Scaling
> > >
> > > We further present empirical results to support our claims using the wall-clock times of the results in our main paper.
> > >
> > > | Dataset (Sec/Epoch) | CE | Peacock |
> > > |---|---|---|
> > > | CIFAR10 | 22.5 ± 0.5 | 22.4 ± 0.8 |
> > > | CIFAR100 | 50.0 ± 0.7 |  50.5 ± 0.2 |
> > > | TinyImageNet | 237.2 ± 1.2 | 240.7 ± 3.6 |
> > > | Camelyon17 | 158.0 ± 3.0 | 159.0 ± 1.5 |
> > > | iWildCam |395.0 ± 2.5 | 392.0 ± 3.5 |
> > > | FmoW | 107.5 ± 1.5 | 110.2 ± 2.0 |
> > >
> > > Our findings demonstrate that Peacock's performance is unaffected by dataset size, maintaining comparable computational efficiency to cross-entropy loss across diverse datasets. These results will be reflected in our main paper. Due to the significant size of ImageNet (134GB), we plan to train on this dataset once we have access to more computational resources. However, we believe that our current experiments provide sufficient evidence to support our conclusions.
> > >
> > > > Post-hoc calibration experiments
> > >
> > > We wish to clarify a misunderstanding regarding this. As our research focuses on out-of-distribution shifts, the results presented in [1] are not directly applicable. To this end, we have integrated the ensemble temperature scaling (ETS) method proposed in [1] into our new experimental results on CIFAR100 and CIFAR100C:
> > >
> > > | Method (CIFAR100) | ECE |
> > > |---|---|
> > > | CE | 5.4_$\pm{0.5}$ |
> > > | CE + ETS | 5.3_$\pm{0.5}$ |
> > > | Peacock Eq. | 4.1_$\pm{0.3}$ |
> > > | Peacock Eq. + ETS | 3.7_$\pm{0.3}$ |
> > > | Peacock Attn. | **3.9**_$\pm{0.2}$ |
> > > | Peacock Attn. + ETS | **2.8**_$\pm{0.2}$ |
> > >
> > > The following table is further evaluated using ETS on out-of-distribution shift CIFAR100.
> > >
> > > |Method (CIFAR100-C) | ECE |
> > > |---|---|
> > > | CE | 10.6_$\pm{0.1}$ |
> > > | CE + ETS | 11.2_$\pm{0.7}$ |
> > > | Peacock Eq. | 9.6_$\pm{0.2}$ |
> > > | Peacock Eq. + ETS | 8.3_$\pm{0.6}$ |
> > > | Peacock Attn. | **9.3**_$\pm{0.1}$ |
> > > | Peacock Attn. + ETS | **7.5**_$\pm{0.1}$ |
> > >
> > > On both ID and OOD test set, we show that Peacock achieves the best results. These additional experiments with ETS demonstrates that our claims are **consistent with the results in our main paper and previous rebuttal**. We thank the reviewer for these suggestions and will incorporate the complete results of these findings into our paper.

---

> > > ### Author Response · Authors · 2024-12-01
> > > **Kindly let us know if you have anymore questions**
> > >
> > > Dear Reviewer jFTV,
> > >
> > > Thank you again for your review and comments. Since the discussion period is closing soon, we are wondering whether our responses have addressed your concerns. Otherwise, we would be happy to continue the discussion and provide further clarifications or explanations.
> > >
> > > Sincerely,
> > > Authors

---

### Official Review · Reviewer_QmtL · 2024-11-01

**Soundness:** 3
**Presentation:** 3
**Contribution:** 3
**Rating:** 6
**Confidence:** 4

**Summary:**

This paper introduces a framework (Peacock) designed to improve neural network calibration through multi-objective optimization. Emphasizing the importance of accurate model confidence in safety-critical applications, the framework proposes combining multiple calibration methods into a single framework. Peacock optimizes these methods to reduce calibration errors for in-distribution (ID) and OOD data. They operate in two modes: Equal Importance, where all methods are weighted equally, and Weighted Importance, where weights are dynamically adjusted using a self-attention mechanism. The results show that Peacock achieves better calibration accuracy across various datasets than any individual method.

**Strengths:**

1. Peacock’s approach to combining calibration methods in a multi-objective framework is novel and reflects a strong and thoughtful attempt to address the limitations of single-method approaches.
2. Provide a theoretical foundation, arguing that Peacock’s calibration error is bounded by the average error of its components. This mathematical rigor adds to the credibility and applicability of their approach however I need to see other reviewers’ opinion on it.
3. Peacock’s use of a self-attention-based weighting mechanism in its Weighted Importance formulation is a valuable addition, allowing the framework to adapt to different calibration needs.
4. The extensive evaluation on synthetic and real-world datasets, including CIFAR variants and iWildCam, demonstrates Peacock’s potential for generalizable performance in ID and OOD contexts.

**Weaknesses:**

1. The work could improve clarity around why certain calibration methods were selected and how they interact synergistically within Peacock. It would be helpful if the authors compared Peacock with other excluded methods to understand its compatibility and potential flexibility.
2. While the self-attention mechanism for adaptive weighting is innovative, the authors do not provide a clear explanation of how specific weights are adjusted in various calibration scenarios. Detailed case studies or visualizations could strengthen the interpretability of the model’s weighting adjustments.
3. Peacock’s performance is documented on many typical benchmarks, but it remains unclear if it scales efficiently in more resource-constrained settings (less human supervision specially) or real-time applications. Details on Peacock’s resource usage across varying computational capacities could improve the overall proposal of framework.
4. The dependence on Expected Calibration Error (ECE) and related metrics is standard, yet calibration results could benefit from additional metrics or more especially contextual tests. It would be great to see the limitation of Peacock in unusual or extreme OOD scenarios.

**Questions:**

1. Please clarify the selection of these specific calibration techniques. How does Peacock handle any dependencies or conflicting effects among these methods, and what might its performance look like if additional calibration techniques (e.g., ensemble-based or regularization-based) were included? It will be helpful to justify the proposed framework and provide comprehensive details for future research.
2. Interpretability of Adaptive Weighting: For the self-attention mechanism, can the authors provide insights into how weight adjustments occur across different datasets? Examples of how specific weights change or stay consistent across OOD shifts would clarify the mechanism’s adaptability.
3. Handling of Adversarial and Non-Standard OOD Scenarios: Peacock performs well in ID and standard OOD settings, but could the authors elaborate on its behavior in more adversarial or highly unstructured OOD contexts? Even very limited empirical evidence will be suffice for it.

---

> ### Author Response · Authors · 2024-11-23
> **Rebuttal**
>
> ### Reviewer QmtL
> We thank the reviewer for the thoughtful and insightful questions. We address the following concerns below:
>
> >Q: Why certain calibration methods were selected and how they interact synergistically within Peacock
>
> 1. A key selection criteria for specific calibration methods was based on a careful consideration of their recent contributions to the field. These techniques tend to focus on prevalent issues such as overconfidence and miscalibration under OOD shifts, which are addressed with entropy methods such as AdaFocal, MaxEnt etc.
>
> 2. In terms of compatibility, these methods can synergistically enhance each other. For instance, both Dual and MaxEnt employ the focal hyperparameter $\gamma$. Rather than assigning a fixed value, we can adaptively obtain $\gamma$ using the AdaFocal method. In another example, RankMixup aligns model confidences based on the ordinal relationships of mixed images, this additionally allows the model to be further trained on interpolated samples.
>
> By carefully selecting and combining these recent contributions, calibration performance can be improved across a variety of tasks and datasets.
>
> >Q: Interpretability of Adaptive Weighting
>
> Our direction weighted self-attention block is designed to balance the learning of different calibration losses. This is due to Equation 16, which assigns smaller weights to bigger losses and larger weights to smaller losses, which is aimed at balancing the learning of loss terms during training. We have included an additional discussion illustrating the learning dynamics of the self-attention block in Page 23 (Appendix B.3) of our manuscript.
>
> >Q: Handling of Adversarial and Non-Standard OOD Scenarios
>
> We thank the reviewer for this interesting question and included additional experiements evaluating all methods on CIFAR100/C.
> We evaluated all methods under Fast-Gradient Sign Method (FGSM) attack, using a standard epsilon value of 0.001. There are two settings to these tables ID + FGSM and OOD + FGSM. For both cases we observe a noticeable drop in model accuracy and increase ECE. However, both variants of Peacock remain relatively well-calibrated with the importance variant performing best. While our initial goal was to tackle OOD shifts, Peacock's performance against adversarial attacks is a significant and unexpected benefit.  We plan to expland on these results and further explore this in future research.
>
> > In-Distribution + FGSM adversarial attack
> | Metric | CE | MaxEnt | AdaFocal | RMixup | CPC | Dual | ACLS | Peacock (Eq.) | Peacock (Impt.) |
> |---|---|---|---|---|---|---|---|---|---|
> |Acc. |74.7_$\pm{0.2}$|71.7_$\pm{0.2}$|75.2_$\pm{0.3}$|74.4_$\pm{0.3}$|74.2_$\pm{0.1}$|74.6_$\pm{0.2}$|74.8_$\pm{0.1}$|74.2_$\pm{0.2}$|74.3_$\pm{0.2}$|
> |ECE|5.7_$\pm{0.2}$|4.9_$\pm{0.2}$|6.6_$\pm{0.2}$|5.1_$\pm{0.2}$|9.3_$\pm{0.2}$|8.5_$\pm{0.2}$|4.8_$\pm{0.2}$| **4.7**_$\pm{0.2}$$|**3.7**_$\pm{0.2}$|
>
> > Out-of-Distribution + FGSM adversarial attack
> | Metric | CE | MaxEnt | AdaFocal | RMixup | CPC | Dual | ACLS | Peacock (Eq.) | Peacock (Impt.) |
> |---|---|---|---|---|---|---|---|---|---|
> |Acc. |51.7_$\pm{0.1}$|52.0_$\pm{0.3}$|52.6_$\pm{0.2}$|52.4_$\pm{0.1}$|51.5_$\pm{0.1}$|51.7_$\pm{0.2}$|51.8_$\pm{0.1}$|51.3_$\pm{0.3}$|51.2_$\pm{0.1}$|
> |ECE|10.1_$\pm{0.1}$|10.8_$\pm{0.3}$|13.5_$\pm{0.2}$|8.6_$\pm{0.2}$|13.7_$\pm{0.1}$|15.2_$\pm{0.2}$|10.2_$\pm{0.1}$|**9.9**_$\pm{0.3}$| **9.5**_$\pm{0.1}$|
>
> We truly thank the reviewer for these insights and are committed to improving our work and addressing any remaining concerns.

---

> ### Comment · Reviewer_QmtL · 2024-11-25
> **Feedback and follow up on author's response**
>
> 1. The response to the question on adaptive weighting (Q2) lacks sufficient scientific justification and empirical evidence. The explanation provided merely reiterates general claims about the benefits of adaptiveness without offering rigorous theoretical backing or detailed experimental results. The question specifically sought to understand why adaptiveness is necessary when simpler heuristic such as assigning smaller weights to larger loss terms and vice-versa can achieve similar goals. However, the response does not focus how the proposed mechanism can offer a tangible advantage over heuristic-based rules.  either a theoretical property or performance comparisons was expected here to validate its effectiveness in the Peacock framework.
>
> 2. The claim that Peacock is a general calibration framework appears to be overstated, as all the empirical evidence provided is focused solely on classification tasks. Although the authors suggest that the framework could potentially generalize to other tasks, they offer no theoretical justification or experimental results to support this assertion. This limitation effectively restricts the framework's applicability to classification tasks, particularly in OOD settings, which contradicts the broader scope implied in the paper. Moreover, demonstrating generalization to non-classification tasks would likely require additional empirical studies and new formulations, which go beyond the rule of conference. Without solid theoretical backing as only the option, the claims regarding the framework's versatility making the position of proposed work weaker.
>
> 3. Proof provided in Appendix C1, while insightful, does not adequately address practical scenarios where calibration losses interact or conflict. The assumptions underlying the bound's validity may not hold in diverse settings, particularly under OOD shifts or adversarial conditions.
>
> 4. Several weaknesses highlighted in the initial review remain unaddressed, such as the conditions for equality in the bound and the implications of conflicting gradients. It is also important to note that the responses were received very late, leaving insufficient time for deeper discussion. While the framework shows promise, further evidence is required to demonstrate its robustness under adversarial performance scenarios. Therefore, I am maintaining the same score and emphasize that these queries need to be fully addressed to justify the claims made in the submission.
>
> I would like to highlight while adversarial attack scenario related investigation is decent but providing rebuttal response quite lately does not give much scope to have detailed discussion towards more informed decision. It could be handled more efficiently.
>
> I am open to read the response for given still in given deadline to give my final score.

---

> > ### Author Response · Authors · 2024-11-25
> > **Rebuttal (cont.)**
> >
> > ### Reviewer QmtL
> > We truly thank the reviewer for responding to our rebuttal, here we seek to provide the following clarifications on our work:
> >
> > >Q: Adaptive weighting (Q2) lacks sufficient scientific justification and empirical evidence.
> >
> > 1. Theoretical support:
> > * To balance the learning of losses, we build our work upon MT-MOO[1], which uses a seperate optimizer for the weight assignments of different losses. This step is achieved in Equation 15, where previous works have shown that solutions for Equation 15 provides a gradient direction that can improve all loss components. (L338-L339). While MT-MOO is largely successful, it can be slow. Specifically recomputing $ \nabla_\theta \mathcal{L}_t(\theta)$ can be expensive as it requires the retention of the computational graph for $A$ backward passes.
> >
> > * To address this issue, our work therefore aims to close this gap in Equation 16, which approximates the loss gradients with their decrease rate estimates $\sqrt{ \frac{\Delta_{\boldsymbol{\theta}} \mathcal{L}_t(\boldsymbol{\theta}) }{ \eta}}$. Appendix C2 provides a rigourous justification for this approximation, including a simple empirical example. Finally, our Direction Weighted Self-Attention block is then optimized using Equation 16, which accepts a vector containing loss terms and outputs a set of weights during training. All the key steps of Peacock are given in Algorithm 1. The learning of weights is illustrated in Figure 7, which shows how our importance weights "control" the learning of losses as compared to assigning uniform weights.
> >
> > 2. Empirical support:
> > * We compare our method against four other multi-objective optimization baselines and the equal formulation in Table 3 and Figure 5. Specifically, we compare our adaptive weighting method against MTAN, CoVV, GradNorm and MT-MOO. Empirically, we can see that our method closely follows MT-MOO, with similar results but is faster. While we currently do not have any justification on our choice of using self-attention, it is currently a prevalent technique in transformers making it a natural choice for our work.
> >
> > 3. Simpler heuristics:
> > * While simple heuristics like assigning smaller weights to larger loss terms can be effective in some cases, it can be difficult to decide on the exact weighting scheme. To resolve this issue, a common approach in deep learning would be to assign loss weights based on their gradients [1][2]. Our adaptive weighting method is close to this aspect, which leverages the decrease rate estimates instead of the gradients to dynamically adjust the weights. Together with the self-attention block, this is something new that we are proposing and welcome feedback by the reviewer.
> >
> > >Q: General calibration framework
> >
> > While we acknowledge that the focus of our work is based on classification tasks, we believe that this is a good way to focus on the issue of miscalibration. The core principle in calibration is based on aligning the model's confidence with it's correctness (L35-L37). This is crucial for models deployed in the real world, such as credit fraud detection, cancer detection etc, as decisions are made not only by the predictions but also judged based on the model's confidence. We understand reviewer's concerns given that calibration is a niche topic, however, we find our experiments on classification tasks to be adequate and in line with recent publications in the calibration literature [3][4][5][6]. Our original scope for this paper was aimed at ID + OOD calibration, which we have extensively evaluated across nine algorithms on eight benchmarks. As for the additional results on adversarial settings, we have deeply thanked the reviewer in our previous rebuttal and are inclined to investigate these results for other tasks in future work.
> >
> > >Q: Proof and bounds
> >
> > A key contribution of this paper is to empirically demonstrate the benefits of jointly optimizing multiple calibration loss functions. The proofs in Sec 4.1 and Appendix C.1 are used to theoretically support our motivation, with our experiments empirically demonstrating this. While the error bounds in diverse settings may not seem entirely obvious, they do hold even on OOD shifts and adversarial conditions. This is because the error bounds of Peacock do not rely on any distributional assumptions about the test set, but instead relies on the mathematical properties of the combined loss function. Specifically, even if individual components may slightly degrade OOD, the loss ensemble of Peacock (which includes other methods) helo mitigate these effect, leading to more robust performance.

---

> > ### Author Response · Authors · 2024-11-28
> > **Kindly let us know if you have anymore questions**
> >
> > Dear Reviewer QmtL
> >
> > We thank you again for your valuable time in reviewing. As we believe we have engaged in the review adequately, we are willing to further answer any more questions and concerns in the discussion phase.
> >
> > We thank you again in reviewing and helping us to improve our work.
> >
> > Please kindly let us know your concerns and if further clarifications are needed and we would be more than happy to answer them.
> >
> > Sincerely, Authors

---

> ### Author Response · Authors · 2024-11-25
> **Rebuttal (cont.)**
>
> >Q: Initial review remain unaddressed:
>
> To the best of our ability, we find the key concerns in the initial review to be adequately addressed:
> 1. Component synergy: We choose components based on their latest developments and the specific issues of miscalibration addressed.
> 2. Adaptive weighting: We have supplemented a figure and discussion on how losses are optimized using the equal versus our proposed importance formulation in Appendix B.3. By using our propose direction weighted self-attention and Equation 16, we can assign weights that control the magnitudes of individual loss terms during training.
> 3. Adversarial settings: We have supplemented new results in our previous rebuttal.
>
> >Q: Rebuttal response time:
>
> The late response to the initial review is indeed regrettable. To that, we thank the reviewer for their patience and seek their understanding as the main author was unable to provide a more timely response due to unforeseen events.
>
> We thank the reviewer again for their response and appreciate their thoughtful and valuable feedback in helping us improve our work. We welcome any further questions during the discussion phase.
> * [1] Ozan Sener and Vladlen Koltun. Multi-task learning as multi-objective optimization. In NeurIPS18
> * [2] Zhao Chen, Vijay Badrinarayanan, Chen-Yu Lee, and Andrew Rabinovich. GradNorm: Gradient
> normalization for adaptive loss balancing in deep multitask networks. ICML18
> * [3] Jishnu Mukhoti, Viveka Kulharia, Amartya Sanyal, Stuart Golodetz, Philip Torr, and Puneet Dokania. Calibrating deep neural networks using focal loss. NeurIPS20
> * [4] Jongyoun Noh, Hyekang Park, Junghyup Lee, and Bumsub Ham. Rankmixup: Ranking-based mixup training for network calibration. ICCV23
> * [5] Hyekang Park, Jongyoun Noh, Youngmin Oh, Donghyeon Baek, and Bumsub Ham. Acls: Adap-
> tive and conditional label smoothing for network calibration. ICCV23
> * [6] Dexter Neo, Stefan Winkler, and Tsuhan Chen. Maxent loss: Constrained maximum entropy for calibration under out-of-distribution shift. AAAI24

---

> ### Comment · Reviewer_QmtL · 2024-11-28
> **Final remarks**
>
> I belive the explanations provided by authors, specifically in second round of discussion pushed the quality of paper towards much better and convincing than initial phase. Given the fact that author shall include discussed reasoning clearly in appendix, I decided to increase the score from 5 to 6. Thank you for detailed answers.

---

> > ### Author Response · Authors · 2024-12-03
> > **Thank you**
> >
> > Dear reviewer, we thank you for helping us improve our work and raising your score.
> >
> > Sincerely
> > Authors

---

### Official Review · Reviewer_CrXD · 2024-11-04

**Soundness:** 2
**Presentation:** 3
**Contribution:** 1
**Rating:** 3
**Confidence:** 4

**Summary:**

This paper presents a comprehensive survey on “Multi-objective optimization for deep neural network optimization”. Motivated from the previous literature, this paper advocate a method of unifying the previous baselines into one, dubbed the method “Peacock”.

This method, according to figure 1(a), encompasses a few recent works: AdaFocal, Dual Focal, and MaxEnt for entropy based one, as well as margin-based CPC and regularizers based RankMixup and Post-hoc processing AdaTS.

Experiments are conducted on many domain shift datasets, like CIFAR10/100-C, FmoW, CivilComments and so on.

**Strengths:**

+ This paper presents a complete survey of previous works.
+ It seems to be a good reading material for the newcomers to this field.
+ Experiments shown in this paper indicates averaging some of the good methods can improve the performance further. This is good to be known for the community

**Weaknesses:**

There is no solid scientific contribution in terms of novel methods. Instead, it basically composes several state-of-the-art methods and constitutes as an engineering effort to unify previous methods. I think in general this paper does not have technical algorithmic novelty, and seems to be a technical report rather than a scientific paper.

There exists some presentation mistakes:
- In the figure of page 2, It should be Figure 1, but this caption is missing.
- I am not sure why this method is called “Peacock”, it lacks of any interpretation but seems to be a meaningless shortcut.

**Questions:**

As above.

My biggest concern is this paper is taking a short cut, summarizing many previous works and put them into one in a naive manner. This might take some of the works, but not the actual contribution that excites the community.

---

> ### Author Response · Authors · 2024-11-23
> **Rebuttal**
>
> ### Reviewer CrXD
> We thank the reviewer for their comments and address concerns below:
>
> > Q: Presentation
>
> We have updated Figure 2 to reflect this. We choose Peacock as an umbrella term to showcase different calibration components.
> We understand that the name might not be immediately intuitive, but believe our naming convention is unique and appropriate.
>
> > Q: Meaningless shortcut
>
> We respectfully disagree, without any references we find this comment difficult to respond to. We kindly ask the reviewer, what exactly are the basis of these claims?

---

> > ### Comment · Reviewer_CrXD · 2024-11-26
> > **Alright**
> >
> > I have no further comments but maintaining my score.

---

> > > ### Author Response · Authors · 2024-11-28
> > > **Kindly let us know if you have anymore questions**
> > >
> > > Dear Reviewer CrXD,
> > >
> > > We thank you again for your valuable time in reviewing. As we believe we have engaged in the initial review adequately, we are willing to further answer any more questions in the discussion phase.
> > >
> > > Please kindly let us know if you have anymore concerns and if further clarifications are needed.
> > >
> > > Sincerely,
> > > Authors

---

### Official Review · Reviewer_pwAz · 2024-11-05

**Soundness:** 3
**Presentation:** 3
**Contribution:** 3
**Rating:** 6
**Confidence:** 4

**Summary:**

The paper presents Peacock, a calibration algorithm that performs multi-objective optimization of existing calibration techniques. The method is shown to improve in and out-of-distribution calibration while retaining performance and speed on computer vision and natural language benchmarks.

**Strengths:**

– The paper is very well written and easy to follow

– The paper does a good job of thoroughly reviewing prior work

– The paper includes a comprehensive appendix with extensive details as well as a detailed codebase which will aid in reproducibility

– The theoretical justification for the multi-objective formulation is intuitive and well-explained

– The proposed method seems effective on a range of datasets without adding a computational overhead

**Weaknesses:**

– The paper framing refers to calibrating “deep neural networks” in general rather than classification models. From what I can tell the proposed approach is designed only with classification in mind – it would be helpful to either update the paper’s framing to reflect this, or to include additional discussion (or if possible, even a proof of concept experiment) demonstrating how Peacock may be generalized to other tasks (or even to specialized classification tasks such as dense prediction)

– The technical contribution of the paper is somewhat limited – 1) it shows that multi-objective optimization of existing calibration strategies is both theoretically and empirically better than or equivalent to any one in isolation (which is consistent with related findings in loss and model ensembling, as the paper acknowledges), and 2) it proposes an algorithm to compute “pareto optimal importance weights” efficiently. That said, it does a very thorough job of exploring various design choices, motivating its method, and reviewing the literature, due to which I do not consider this a significant weakness.

– The “weighted importance formulation” would benefit from additional motivation and . L306 claims that “loss terms can often be conflicting” as the motivation for importance weighting – it would be good to experimentally demonstrate this. Further, do any of the theoretical guarantees for the equal weighting formulation apply to the importance weighted version? Finally, why does importance weighting not consistently outperform equal weighting in Table 1 and 2?

– Some aspects of the method design need elaboration. For eg. why is “direction weighted self-attention” required to learn loss weights, rather than say a simple linear layer?

**Questions:**

Please see weaknesses above for a detailed list of questions.

---

> ### Author Response · Authors · 2024-11-23
> **Rebuttal**
>
> ### Reviewer pwAz
> We thank the reviewer for the interesting questions and feedback. We address the following concerns below:
>
> >Q: Paper framing.
>
> The current framing of our paper as "calibrating deep neural networks" is in line with the convention commonly used in the literature [1][2][3][4].
>
> >Q: How Peacock may be generalized to other tasks such as dense prediction.
>
> While most works on neural network calibration are largely focused on classification tasks, we acknowledge that the core principles of Peacock are applicable to a broader range of tasks.
> We believe that with appropriate adaptations, the benefits of Peacock can be extended to other tasks, such as dense prediction. We plan to explore these extensions in future work.
>
> >Q: Technical contributions:
>
> We appreciate the reviewer’s acknowledgment of our thorough exploration of design choices and reviewing of literature. Although the core idea of multi-objective optimization of calibration strategies may seem intuitive, our work offers the following key contributions:
>
> 1. Our paper highlights the underlying issues in deep neural network calibration such as overconfidence and OOD shifts and pays tribute to all recent contributions.
> 2. We are the first work that introduces a multi-objective framework for neural network calibration, with both and equal and importance weighted formulation.
> 3. We propose an efficient algorithm to compute Pareto optimal importance weights, which is crucial for practical deployment.
>
> Our work goes beyond simply combining existing calibration strategies, but also addressing the key issues faced (e.g. compute overhead, learning instability) during combining.
>
> >Q: Weighted Importance Formulation:
>
> We thank the reviewer for their interest in the self-attention block and have included an additional discussion justifying our decision in using a direction weighted self-attention block in Page 23 (Appendix B.3) of our manuscript.
>
> >Q: “loss terms can often be conflicting” – it would be good to experimentally demonstrate this
>
> By using equal weights, loss terms are optimized equally by the model. We will better phrase this sentence.
>
> >Q: Further, do any of the theoretical guarantees for the equal weighting formulation apply to the importance weighted version?
>
> While we do not have formal theoretical guarantees for the importance-weighted version, our empirical results demonstrate that our importance-weighted variant also obeys the bounds in Equation 12.
>
> >Q: Finally, why does importance weighting not consistently outperform equal weighting in Table 1 and 2?
>
> We thank the reviewer for this question. We currently believe that since the importance weights and temperature scaling AdaTS were obtained ID, the type and magnitude of OOD shift in the test set is the main cause of fluctuation in Table 1 and Table 2.
>
> >Q: Why is “direction weighted self-attention” required to learn loss weights, rather than say a simple linear layer?
>
> Our intuition behind using self-attention is similar to that of Transformers. The K, Q, V matrices allow for the learning of complex relationships, with the added benefit of the softmax function which helps fufill the KKt constraints. While it is possible to use a linear layer, we believe that our choice of using self-attention is also a good option.
>
> * [1] Chuan Guo, Geoff Pleiss, Yu Sun, and Kilian Q. Weinberger. On calibration of modern neural networks. ICML17
> * [2] Jishnu Mukhoti, Viveka Kulharia, Amartya Sanyal, Stuart Golodetz, Philip Torr, and Puneet Dokania. Calibrating deep neural networks using focal loss. NeurIPS20
> * [3] Jongyoun Noh, Hyekang Park, Junghyup Lee, and Bumsub Ham. Rankmixup: Ranking-based mixup training for network calibration. ICCV23
> * [4] Dexter Neo, Stefan Winkler, and Tsuhan Chen. Maxent loss: Constrained maximum entropy for calibration under out-of-distribution shift. AAAI24

---

> > ### Comment · Reviewer_pwAz · 2024-11-25
> > **Thank you for the response**
> >
> > The author response has addressed some of my concerns. However I still have reservations about whether the framing is appropriate, especially since the claim is that Peacock is a "unified framework for deep NN calibration" and that works exploring neural network calibration in other (arguably more realistic) tasks have also been published eg. [A]. Moreover, some recent works in calibration do demonstrate generalization across tasks eg. [B]. Overall I am inclined to retain my borderline rating.
> >
> > [A] Wang, Dongdong, Boqing Gong, and Liqiang Wang. "On calibrating semantic segmentation models: analyses and an algorithm." Proceedings of the IEEE/CVF Conference on Computer Vision and Pattern Recognition. 2023.
> >
> > [B] Chattopadhyay, Prithvijit, et al. "AUGCAL: Improving Sim2Real Adaptation by Uncertainty Calibration on Augmented Synthetic Images." International Conference on Learning Representations. International Conference on Learning Representations, 2024.

---

> > > ### Author Response · Authors · 2024-11-26
> > > **Rebuttal and Thank you**
> > >
> > > ### Reviewer pwAz
> > >
> > > We thank the reviewer for their time and efforts in providing constructive feedback, allowing us to improve our work.
> > >
> > > > Q: Framing
> > >
> > > The authors will discuss and think of a better way to frame the paper.
> > >
> > > > Q: Related works
> > >
> > > We thank the reviewer for these related works and agree that there are many other interesting applications of neural network calibration. That said, we plan to investigate this direction in our future work.
> > >
> > > Once again we thank the reviewer for their positive response of our work and welcome any further questions in the discussion phase.

---

### Author Response · Authors · 2024-12-03
**Global Response and Summary of Discussion Phase**

We sincerely thank all reviewers for their valuable feedback and constructive criticism. We also express our gratitude to the ACs and SACs for their time and guidance.

In response to the insightful comments from Reviewers pwAz and QmtL, we have conducted additional experiments to address their concerns regarding the adaptive weighting mechanism of Peacock and its performance under adversarial attacks. These experiments have been included in our rebuttal, and we are pleased to note that Reviewer QmtL has increased their score from 5 to 6.

Additionally, we have addressed the final comments from Reviewer jFTV by conducting further experiments on scalability and post-hoc processing. We believe these experiments adequately address their concerns.

We thank the reviewers again for their insights, which has significantly improved our work. We wish you all a happy holiday season.

---

### Meta-Review · Area_Chair_j6eg · 2024-12-23

**Metareview:**

This paper presents Peacock, a method to improve calibration through a multi-objective optimization viewpoint. The method specifically combines multiple calibration methods by determining a weighting importance for each method via a direction-weighted self-attention block. Results show improved ID and OOD calibration performance across a wide set of benchmarks. The reviewers appreciated the method's comprehensive description of prior works and attempt to unify them, but raised a number of concerns. While the idea of combining methods is intuitive (and in-line with ensembling/other variants for other domains), a number of considerations such as why particular methods were selected, how they interact (and as mentioned by authors, can conflict), and how this ties to both theory and empirical performance (e.g. closeness of results in many cases with equal weighting). Indeed, Additional concerns included issues in the writing and theoretical notations, interpretation/justification of the self-attention based weight adjustment, handling of adversarial/other OOD scenarios, etc. The authors provided a rebuttal, with a good bit of back-and-forth with the reviewers, including some additional experiments and explanations. Reviewer QmtL was overall satisfied with the response, but reviewers pwAz and jFTV expressed significant concerns were remaining.

After considering all of the materials, I recommend not accepting this paper at this time. Overall, there were a large number of concerns expressed by reviewers, and while we appreciate the author's effort in addressing some of them, many remain outstanding. While the idea itself, especially with theoretical motivation, can be interesting, there is not sufficient scientific rigor in the formulation, demonstrating the need for adaptive weighting, its relationship to when it would/would not work given comments about considerations such as conflicts across methods, and ties to implementation (self-attention). On the flip side, the paper could use at least stronger empirical demonstration of superiority especially of the dynamic weighting component and at larger scale settings that are now common. We recommend that the authors address these to strengthen the contributions for a future resubmission.

**Additional Comments On Reviewer Discussion:**

A large number of concerns were raised across the reviewers; some were addressed well (e.g. framing, comparison to post-calibration, etc.), but others that directly affect the paper's contributions remain, as mentioned above.

---

### Decision · Program_Chairs · 2025-01-22

Reject